# Stress-induced ribosome degradation in *Bacillus subtilis* is mediated by the RNase Y-specificity complex

Fabián A. Cornejo [1,4] ✉, Kristina Driller[1,2,4], Rina Ahmed-Begrich [1], Katja Schmidt [1], Michael Jahn [1], Vivekanandan Shanmuganathan [1], Karin Hahnke[1], Florian Kondrot[1], Thomas F. Wulff [1], Sebastian Rämisch[1], Kathirvel Alagesan [1], Emmanuelle Charpentier [1,3] & Kürşad Turgay [1,2] ✉

Limiting ribosome synthesis and activity is crucial for adaptation to stresses, such as heat or nutrient starvation. In *Bacillus subtilis*, this can be achieved through the coordinated action of the alarmones (p)ppGpp and the transcription factor Spx. Here, we performed a genetic screen to identify novel factors that contribute to the heat shock response in *B. subtilis*. We identified the Y-complex, which confers specificity to the endonuclease RNase Y, as a critical player under stress conditions, such as heat or transition into the stationary phase. This protein complex is required for the targeting and processing of diverse RNAs, notably the maturation of mRNAs encoding proteins involved in translation and metabolism. We further demonstrate that the Y-complex and RNase Y initiate the degradation of rRNAs of mature ribosomes, lowering their abundance. We propose that the Y-complex is a regulatory hub that modulates gene expression, adjusts protein synthesis and resource allocation.

Bacteria can sense, adapt to, and survive stressful environmental changes facilitated by various general and specific stress response systems. Proteotoxic stresses, like high temperatures, significantly disrupt protein folding and can lead to the formation of toxic protein aggregates. Chaperones and proteases are expressed as a response due to the inactivation of the transcriptional repressors HrcA and CtsR in *Bacillus subtilis*[1–4]. The activation of the protein quality control (PQC) allows the refolding of un- or misfolded proteins by different chaperone systems or the degradation of irreparable proteins through AAA+ protease complexes like ClpCP[5,6].

In addition to the repairing or removal of protein aggregates, inhibiting translation is crucial for survival during proteotoxic stresses, as it can alleviate the extra load on the PQC system[7–10]. In *Bacillus subtilis*, the nucleotide second messengers guanosine tetraphosphate and guanosine pentaphosphate, collectively referred to as (p)ppGpp,

and the transcription factor Spx independently regulate synthesis and activity of ribosomes during stress[7,11]. Spx is a redox-sensitive heat shock transcription factor that controls the expression of redox chaperones and simultaneously represses transcription of translation-related genes during oxidative and heat shock response[11–14].

The alarmones (p)ppGpp were initially discovered to be synthesized during amino acid starvation, but more recently, it was observed that their synthesis is also induced by protein folding stresses, such as heat or oxidative stress[7]. During heat stress, their accumulation results in increased thermoresistance, inhibition of translation, and a reduction in protein aggregation[7]. (p)ppGpp inhibit translation by binding to specific GTPases (e.g. IF2, RbgA)[15–17]. Furthermore, (p)ppGpp accumulation can lead to decreased transcription of e.g. rRNA and ribosomal protein (r-protein) transcripts by inhibiting GTP biosynthesis and reducing its levels[18,19]. This downregulation of translation is an

[1]Max Planck Unit for the Science of Pathogens, Berlin, Germany. [2]Leibniz Universität Hannover, Institute of Microbiology, Hannover, Germany. [3]Humboldt-Universität zu Berlin, Institute of Biology, Berlin, Germany. [4]These authors contributed equally: Fabián A. Cornejo, Kristina Driller. ✉e-mail: cornejo@mpusp.mpg.de; turgay@mpusp.mpg.de

important component of the response to nutrient scarcity as well as heat shock.

In this study, we aimed to identify new genes or pathways involved in the heat shock response using a genetic screen that reports on fitness. We identified that the Y-complex (RicAFT-complex), which interacts with the endonuclease RNase Y and conveys substrate specificity, plays a role in the heat shock response. We further demonstrate that the Y-complex regulates ribosome levels by initiating ribosome degradation during nutrient depletion or heat shock. Mutants lacking a Y-complex member show increased ribosome levels when transitioning into stationary phase or during heat shock, increasing the burden on the PQC system and, in consequence, protein aggregation during heat stress. These findings not only indicate the importance of RNases during stress response but also emphasize that

controlling the maturation and stability of RNA is an important general mechanism to modify cellular function.

## Results

### The Y-complex is necessary for survival during heat shock

To identify genes that play a role in the heat shock response, we used the BKE single-mutant library[20], which includes a barcoded deletion cassette replacing each non-essential gene in *B. subtilis*. All gene-deletion mutant strains were pooled, grown at 30 °C, and subjected to a mild heat shock at 50 °C during the mid-exponential phase, while a control was maintained at 30 °C. Relative fitness scores for each gene deletion in the pooled population were calculated by quantifying each barcode at different temperatures (Fig. 1A, Supplementary Data 1). The relative fitness of the heat-shocked cultures compared to the cultures

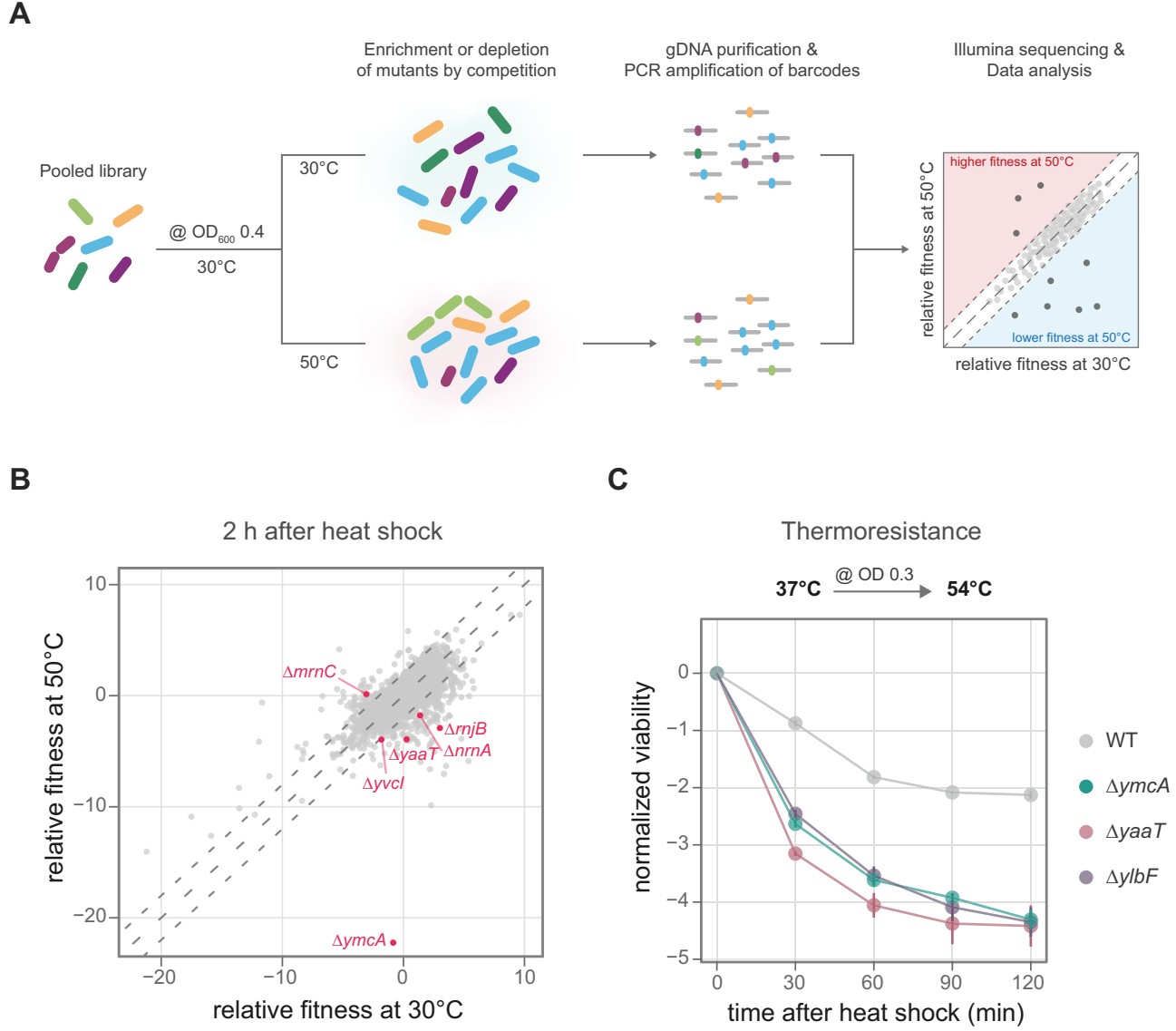

**Fig. 1 | Genetic screening for genes involved in the heat shock response.**
**A** Genetic screening experiment: the pooled library of deletion mutants (BKE) was grown at 30 °C; when it reached an OD_{600nm} of 0.4, one fraction was exposed to 50 °C while the other was maintained at 30 °C. Genomic DNA (gDNA) was extracted from the library, and barcodes were amplified and sequenced using Next-Generation Sequencing. The fitness of each mutant at the mentioned temperatures and time points was calculated using BEANcounter. **B** Relative fitness of each deletion mutant at 50 °C (*y*-axis) and 30 °C (*x*-axis) two hours post-temperature shift. Each point represents a deletion mutant, with mutants lacking RNase-related

genes highlighted in magenta. The dotted lines indicate the range in which the relative fitness between the conditions differed by less than 2. Data represent the average of three biological replicates. A negative relative fitness value means that the abundance of the deletion mutant, compared to the rest of the population, was negatively affected by the treatment, while a positive value means an increased abundance in this condition. **C** Changes in the viability (log_{10}CFU/ml) of WT (grey) and *ymcA* (green)*, ylbF* (purple), and *yaaT* (rose) deletion mutants to a severe heat shock at 54 °C. Values were normalized to the viability at timepoint 0 min. The data represents the average ± standard error of three biological replicates.

grown at 30 °C was used to identify genes whose deletion results in a change in fitness during heat shock (Fig. 1A, B, Supplementary Data 1).

We observed that *B. subtilis* strains lacking the genes encoding the heat shock transcriptional repressors HrcA or CtsR exhibit higher fitness at 50 °C than at 30 °C (Supplementary Fig. 1A), consistent with their known regulatory role[21]. In contrast, deletion of the genes encoding the DnaK chaperone system, the protease ClpP, and the adaptor protein complex McsA-McsB show greater sensitivity to increased temperatures (Supplementary Fig. 1B), demonstrating that our screening approach can identify genes with already known heat-shock phenotypes[22–24].

The control of transcription initiation by transcription factors has been well established[25]. However, the contribution of RNA stability in stress response is not well understood. Interestingly, six mutants lacking RNases or RNase-regulating genes exhibit differential fitness at 50 °C compared to 30 °C (Fig. 1B). The deletion of RNase Mini-III (Δ*mrnC*) results in lower fitness at 30 °C, whereas its fitness remains unaffected at 50 °C. Deletions of RNase J2 (Δ*rnjB),* the nano-RNase A (Δ*nrnA*), and the RNA pyrophosphohydrolase and (p)ppGpp hydrolase NahA (Δ*yvcI*) exhibit reduced fitness during increased temperatures (Fig. 1B).

One of the mutants with the lowest fitness score at 50 °C is Δ*ymcA*. In addition, the related mutant Δ*yaaT* also shows reduced fitness (Fig. 1B). It is well established that YmcA (RicA) interacts with YaaT (RicT) and YlbF (RicF) to form the Y-complex, which associates with the endonuclease RNase Y for maturation of polycistronic mRNA and riboswitches[26,27]. We could confirm the heat sensitivity of all three deletion mutants of the Y-complex genes in single-culture experiments during a severe heat shock at 54 °C (Fig. 1C). In addition, the heat shock sensitivity of the Δ*ymcA* strain can be rescued by ectopically expressing *ymcA* from the *amyE* locus, even without inducer, due to the known leakage of the *hyperspank* promoter[28] (Supplementary Fig. 1C).

### Identification of Y-complex-dependent cleavages at late exponential and transition phases

Interestingly, all three strains lacking the Y-complex encoding genes exhibit a similar growth defect that begins at the end of the exponential phase and persists through the transition to the stationary phase (Fig. 2A). This growth phenotype of the Δ*ymcA* strain is completely rescued in a *ymcA*-complemented strain (Supplementary Fig. 1D). Such a post-exponential growth change in the transition to the stationary phase can be considered a physiological stress where nutrients are depleted slowly from the media. Thereby, activating the general stress response and stringent response, including the CodY regulon[29–31].

Given that this complex associates with the endonuclease RNase Y and regulates substrate selection[27], we investigated Y-complex-dependent cleavages in the late exponential and transition phase (Fig. 2A) using the ISCP (Identification of Specific Cleavage Positions) approach[32]. We compared the coverage of RNA 3′ and 5′ ends in the WT and Δ*ymcA* mutant, where the significantly reduced ends in the Δ*ymcA* were classified as putative Y-complex-dependent cleavages (Supplementary Data 2). It is worth noting that deleting any member of the Y-complex results in a significant decrease in the processing of substrate RNA[27], making the effects observed in the Δ*ymcA* a proxy for the whole Y-complex function. When a transcript is cleaved, it results in two products: one is quickly degraded by exonucleases, while the other is stabilized[32]. Consequently, it is not always possible to detect both 5′ and 3′ ends after cleavage.

We identified 221 and 258 Y-complex-dependent ends in late exponential and transition phases, including six ends previously detected in mid-exponential phase[27] (Supplementary Data 3). In the late exponential phase, 5′ and 3′ ends are almost equally distributed, yet more 5′ ends are detected in the transition phase (Fig. 2B).

It is known that the Y-complex and RNase Y generate fragments with differential stability in the mid-exponential phase[27]. Therefore, we

evaluated the stability of the resulting fragments in the late exponential and transition phases by calculating the ratio of coverage upstream and downstream of the cleavage site (Supplementary Data 2). We observed a trend toward stabilizing downstream fragments. However, upstream fragments can also be stabilized, but to a lesser extent (Supplementary Fig. 2A). The ISCP method differentiates processing sites with blunt ends (Unique, U-type) or stepped (S-type), when it was further processed by exonucleases[32]. Most S-types were found at the 3′ ends, suggesting further processing by exonucleases, while most U-types were detected at 5′ ends (Fig. 2C). This is consistent with the observed trend to stabilise the downstream fragment (Supplementary Fig. 2A).

The Y-complex could target a particular sequence motif, thereby helping to recognize specific substrates. However, a motif search using U-type ends resembles the simple motif described for RNase Y in *B. subtilis*[33], *S. aureus*[34], and *S. pyogenes*[32], being most likely a G upstream of the cleavage site (Supplementary Fig. 2B). Given the simple sequence motif, we investigated whether cleavages occur randomly in longer or more abundant transcripts. Notably, approximately 81% of targeted transcripts are cleaved only once, about 16% twice, and very few are cleaved multiple times (Supplementary Fig. 2C). Neither the length nor the abundance of a transcript predicts whether it is cleaved, suggesting that other factors influence where cleavages occur (Supplementary Fig. 2D–E, Supplementary Data 4). Recent studies have shown that the Y-complex-dependent RNase Y cleavage efficiency is multifactorial, depending on the sequence and secondary structures surrounding the cleavage site[33,35].

As we observed that RNA processing could result in fragments with different stabilities (Supplementary Fig. 2A), we wondered whether this might be due to secondary structures protecting them from exonucleases. We used U-type ends and compared transcripts with the same cleavage motif but not cleaved as controls. For the 5′ ends, a strong decrease in the minimum free energy ~25 nt downstream of the cleavage site indicated a possible secondary structure, while the upstream fragment remained similar to the control (Fig. 2D). The upstream fragment's lower stability may be due to its lack of secondary structure, making it accessible to 3′–5′ exonucleases. For the 3′ ends, the more stable upstream fragment could form a possible secondary structure starting 10 nt upstream of the cleavage site (Fig. 2D). The generation of mRNA fragments with different stability might have a regulatory role in gene expression.

### Y-complex-mediated transcript cleavages and mRNA maturation

Most detected Y-complex-dependent ends are inside ORFs, mainly in polycistronic mRNAs (Fig. 2E–F), confirming the role of the Y-complex in the maturation and stoichiometry regulation of co-transcribed genes[27]. An example is the *cggR-gapA* transcript, where the Y-complex mediates cleavage in the coding sequence of *cggR* near the stop codon[26,27]. Cleavage renders the *cggR* fragment less stable compared to the *gapA* fragment (Supplementary Fig. 3A), thereby uncoupling the transcript levels of these co-transcribed genes. Interestingly, we observed other cleavages generating similar uncoupling of transcripts, such as in the gene encoding the endonuclease YloC (Supplementary Fig. 3B, Supplementary Data 2). Furthermore, cleavages in genes transcribed as monocistronic mRNAs, such as *guaB*, suggest that the Y-complex could initiate their degradation (Supplementary Fig. 3C, Supplementary Data 2).

We observe Y complex-dependent processing in 5′, 3′, and internal UTRs (Fig. 2F). Cleavages in 5′ UTRs suggest a possible regulatory role by decreasing the abundance of the mRNA containing them, as already described for riboswitches[27,36], or by producing a shorter version of a 5′ UTR, which could affect gene expression. Processing of 3′ and internal UTRs could influence mRNA stability and polycistronic mRNA maturation. We observed that some 3′ UTR cleavages are located

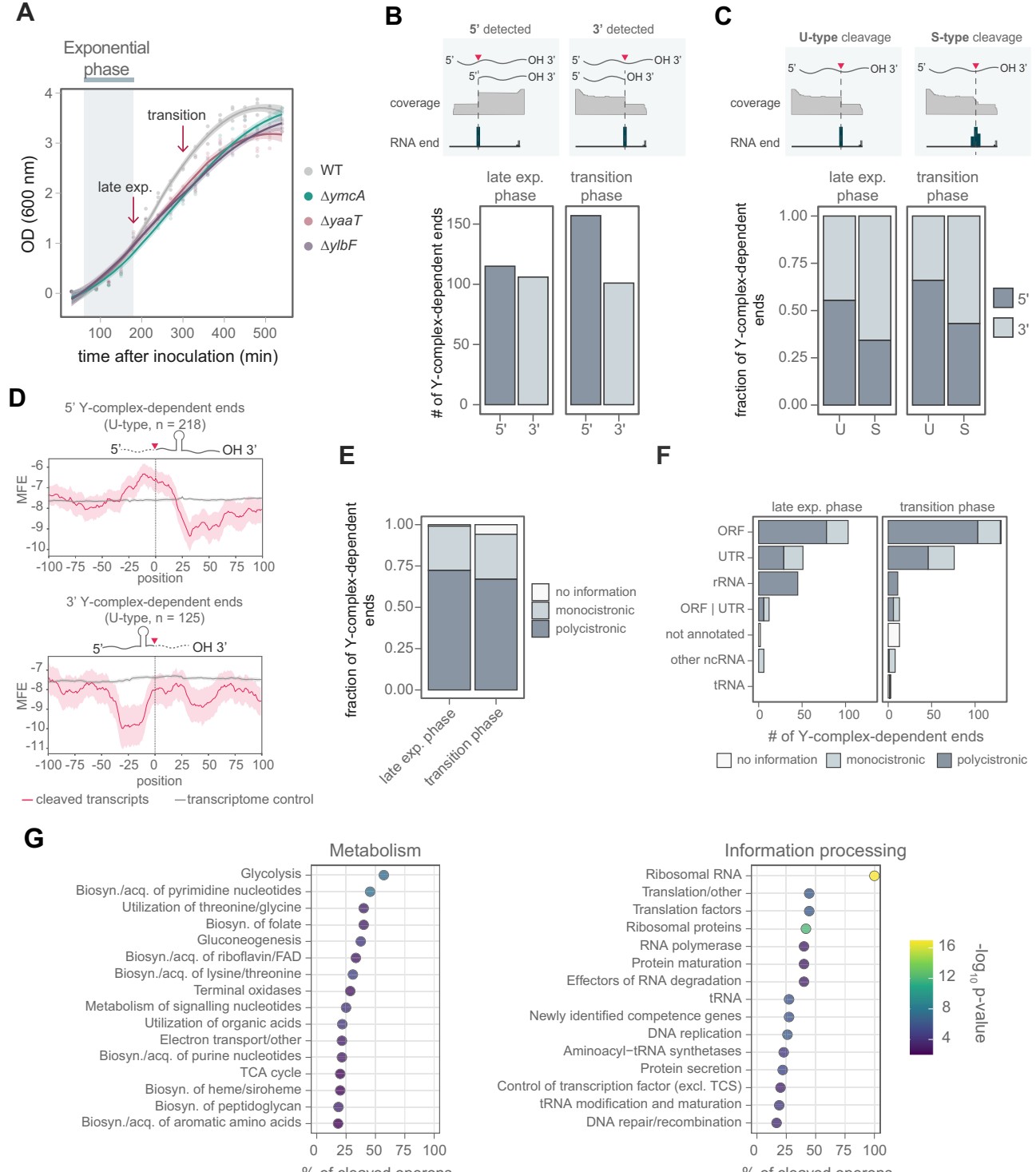

**Fig. 2 | Identification of Y-complex-dependent cleavages at late exponential and transition phase. A** Growth curves of WT (grey), *ymcA* (green), *ylbF* (purple), and *yaaT* (rose) deletion mutants in LB at 37 °C. The blue bar marks the exponential phase. Arrows indicate sample collection for the ISCP method at the late exponential and transition phases. Three biological replicates are displayed with a smooth curve regression (LOESS) and a band representing the 95% confidence interval. The growth phases were classified using a semi-log plot. **B** Number of RNA 3' or 5' ends identified at the two growth phases. **C** Fraction of identified 3' or 5' ends displaying U-type (unique) or S-type (stepped) profiles. **D** Minimum free energy (MFE) was calculated on 50 nt windows for RNAs cleaved by the Y-complex with U-type ends (in magenta). Transcripts that share the same cleavage motif but are not processed were used as a control (in gray). rRNA genes were excluded from the analysis since they are highly structured. The colored band represents the 95%

confidence interval. **E** Fractions of ends identified in genes transcribed in polycistronic or monocistronic RNA. **F** Count of RNA features where Y-complex-dependent ends are detected at either growth phase. Annotations were obtained from BSGatlas[46]. **G** Overrepresentation analysis of pathways whose transcripts are cleaved by the Y-complex during the late exponential phase. Operons were assigned to be involved in a pathway if they code for one gene participating in such a pathway. As genes in the same operon can have different functions, transcripts may be linked to multiple roles. Pathway annotations were retrieved from SubtiWiki[48]. The color indicates the statistical significance (*p*-value) of the enrichment calculated using a hypergeometric test. *p*-values were adjusted for multiple testing using the Benjamini-Hochberg method. Biosyn biosynthesis, acq acquisition, excl TCS excluding two-component systems.

upstream of the Rho-independent terminator, decreasing the stability of the upstream fragment and leaving an apparently more stable downstream fragment that includes its terminator structure, as observed for *trxB* (Supplementary Fig. 3D). This phenomenon has been observed in *E. coli*, where RNase E cleaves upstream of the Rho-independent terminator[37,38].

In polycistronic mRNA maturation, we observed that Y-complex processes internal UTRs, decoupling the RNA levels of coding fragments (Supplementary Fig. 3E–F). For example, the *bipA-ylaH* transcript is cleaved 21 nt after the *bipA* stop codon, rendering the upstream fragment less stable than the *ylaH* fragment (Supplementary Fig. 3E). In addition, we observed a cleavage between the *fusA* (EF-G) and *tufA* (EF-Tu) genes, both coding for translation elongation factors (Supplementary Fig. 3F).

## The Y-complex targets translation and metabolism-related genes in the late exponential phase

A specific cleavage motif of the Y-complex could not be identified. Therefore, we wondered whether this multifactorial cleavage motif has evolved in transcripts with specific functions required to switch the growth phase. We conducted an operon-based overrepresentation analysis to determine whether the target transcripts of the Y-complex encode genes with specific functions. We observed that the most significantly enriched categories pertain to "Metabolism" and "Information processing" (Fig. 2G, Supplementary Fig. 4A, Supplementary Data 5). Overall, the Y-complex is predominantly involved in the processing of operons encoding genes for carbon metabolism (e.g., glycolysis and gluconeogenesis), nucleotide metabolism (e.g., pyrimidine and purine biosynthesis), and amino acid biosynthesis and utilization (Fig. 2G). This allows adjusting the utilization and biosynthesis of nutrients when they become limited. In addition, all transcripts containing rRNA and ~50% of transcripts carrying genes encoding translation factors and r-proteins are processed by the Y-complex (Fig. 2G).

The reduced RNA processing in the *ymcA* deletion strain results in significant changes in the transcriptome (Supplementary Fig. 4B, Supplementary Data 6–7). Consequently, transcripts of certain pathways related to metabolism and translation, among others, show maturation events that depend on the Y-complex and also demonstrate differential expression upon deletion of *ymcA* (Fig. 2G, Supplementary Fig. 4B). This may result from either a direct or indirect effect of decreased RNA maturation events. Our results suggest that the Y-complex and RNase Y are important for transitioning to the stationary phase by processing transcripts related to translation and metabolism, which may impact their stability and/or gene expression.

## The Y-complex targets ribosomes for degradation

Several Y complex-dependent cleavages were detected in the rRNA sequence during both growth phases, most of them being in the 23S rRNA during the late exponential phase (Fig. 3A). We detected cleavages in specific helices in the 23S and 16S rRNA, mainly in loops of hairpin structures (Supplementary Fig. 5A–B). These helices are accessible and primarily located close to the A-site and the interface between the two ribosomal subunits (Supplementary Fig. 6), which could be recognized and cut by the Y-complex with the endonuclease RNase Y. These cleavages will most likely destabilize the ribosome structure. Generating free 5′ and 3′ RNA ends, which are substrates for exonucleases such as PNPase, RNase R, and RNase J, will result in further RNA degradation and ribosome disassembly, releasing the ribosomal proteins as well.

We confirmed the observed rRNA degradation via northern blots of 23S or 16S rRNA (Fig. 3B, Supplementary Fig. 7A). Deletion of *ymcA* strongly reduces the presence of rRNA degradation intermediates during the late exponential phase. As ribosomes are assembled co-transcriptionally[39], we inhibited transcription by adding rifampicin to cultures at the late exponential phase. We observed that stopping transcription does not affect rRNA degradation of the 23S and even raises the levels of 16S degradation intermediates, indicating that mature ribosomes are degraded (Fig. 3B, Supplementary Fig. 7A). Given that the Y-complex protein YaaT interacts more stably with RNase Y in the membrane than YmcA and YlbF[40], we tagged YaaT with a C-terminal FLAG to identify interacting proteins during the late exponential phase (Supplementary Fig. 7B, Supplementary Data 8). We observed that YaaT captures other Y-complex members (YmcA, YlbF), RNase Y, and components of the dynamic degradosome complex, including enolase and PNPase. Interestingly, r-proteins are also observed in the pull-down experiment, suggesting the Y-complex might target ribosomes to RNase Y for initiating rRNA degradation after exponential growth (Supplementary Fig. 7B).

To confirm the involvement of RNase Y in rRNA degradation, we constructed a conditional depletion strain in which the single copy of *rny* (coding for RNase Y) is under the control of a xylose-inducible promoter. Depletion of RNase Y results in a growth defect during the transition phase, mirroring what we observed with the Y-complex mutants (Supplementary Fig. 7C). The degradation of 23S rRNA is highly impaired by the depletion of RNase Y (Supplementary Fig. 7D), confirming the involvement of RNase Y together with the Y-complex in ribosome degradation.

Cleavages within mature rRNA can trigger ribosome disassembly and degradation, which, during the transition phase, coincides with translation downregulation through reduced synthesis and inhibition of ribosome activity. This suggests that the Y-complex may be involved in reducing ribosome levels to accommodate the new, slower growth rate. Therefore, we quantified ribosome content by measuring the abundance of all r-proteins via mass spectrometry (Supplementary Data 9). When *B. subtilis* cells shift from exponential to the transition phase, we observed a coordinated reduction in r-protein levels (Fig. 3C), indicating a decrease in ribosome levels. This reduction is highly impaired in Δ*ymcA* (Fig. 3C), where the levels of r-proteins do not decline to the same degree as in the WT, likely due to the reduced degradation of ribosomes.

Accumulation of (p)ppGpp leads to inhibition of transcription of rRNA and r-proteins[19,41] upon shortages of nutrients during the transition to the stationary phase. To test if the deletion of *ymcA* is interfering with this transcriptional downregulation, we measured the promoter activity of the rRNA promoters *rrnJ* and *rrnB* and r-protein promoters of *rplK*, *rplJ*, and *rpsJ*. All these promoters show similar downregulation in the late exponential and transition phases in both WT and Δ*ymcA* strains (Fig. 3D-E, Supplementary Fig. 7E–G), confirming that their transcriptional control remains unaffected and is not the cause of the increased ribosome amounts in the Δ*ymcA* strain.

To distinguish active degradation of ribosomes from changes in their biogenesis, we quantified ribosome turnover in vivo using an inducible uL1-HaloTag fusion to label the 50S subunit. Ribosomes were pulse-labelled with a fluorescent TMR-ligand during the exponential phase and subsequently chased in conditioned media after blocking further labelling with 7-bromo-1-heptanol[42] and analyzed by microscopy (Fig. 3F). Ribosome half-life more than doubled in the Δ*ymcA* mutant ($51.2 \pm 10.7$ min) compared to the wild-type ($22.8 \pm 1.08$ min) during the transition to stationary phase (Fig. 3G). These results suggest that the Y-complex is required for the active turnover of mature ribosomes, confirming that the observed accumulation of ribosomes in the Δ*ymcA* is driven by reduced degradation rather than increased ribosome biogenesis.

Notably, we observed that the increased ribosome levels in a Δ*ymcA* strain led to the accumulation of hibernating 100S ribosomes during the transition phase (Fig. 3H). These 100S ribosomes are no longer observed after the deletion of the gene encoding the hibernation factor *hpf*, which triggers their formation[43,44] (Supplementary Fig. 7H). The presence of hibernating ribosomes at this time point indicates that functional and highly abundant ribosomes in the Δ*ymcA*

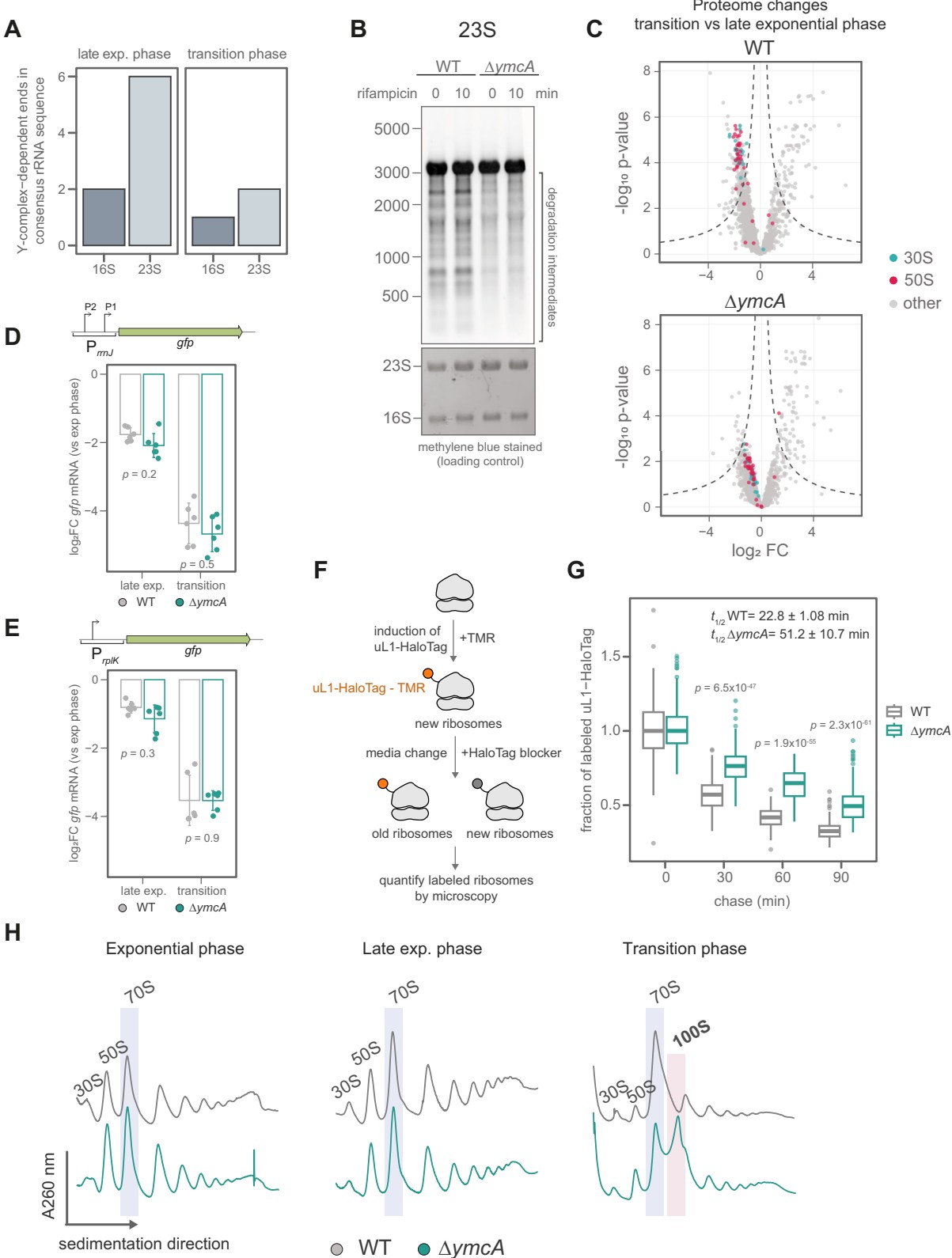

strain are sequestered from the actively translating ribosome pool through hibernation.

### (p)ppGpp and the Y-complex play a crucial role in regulating the levels of ribosomes

Since ribosome levels and translation are lowered by (p)ppGpp under nutrient limitation or other stresses, we examined whether a genetic interaction exists between the two systems. Deleting *ymcA* in a strain that cannot synthesize (p)ppGpp [(p)ppGpp⁰] leads to an even stronger growth defect during the transition phase (Fig. 4A). The (p)ppGpp⁰ strain, like Δ*ymcA*, shows impaired reduction of ribosome levels during the transition phase (Fig. 4B). The double mutant (p)ppGpp⁰ Δ*ymcA* shows almost no reduction in ribosome quantities during this growth phase (Fig. 4B–C). The same is

**Fig. 3 | The Y-complex controls ribosome levels by initiating rRNA degradation.** **A** Number of cleavages detected by ISCP in *B. subtilis* rRNA during the late expo-nential and transition phase. **B** Northern blot against 23S of WT and Δ*ymcA* cells at the late exponential phase, before and after treatment with 100 μg/ml rifampicin to stop transcription. The methylene blue-stained membrane is shown as a loading control. This image is representative of three biological replicates. **C** Volcano plot of protein quantification of WT and Δ*ymcA* strains comparing transition to late exponential phase. The proteins composing the small (30S) and major (50S) ribo-some subunits are highlighted in cyan and magenta, respectively. The data shows the average of four biological replicates. The *p*-values were calculated using a two-sided Student's *t*-test and adjusted using the Benjamini-Hochberg method. Activity of the promoter controlling (**D**) *rrnJ* (rRNA) or (**E**) the *rplK* (r-protein) transcription at late exponential and transition phase compared to the exponential phase in WT (grey) and Δ*ymcA* (green). The promoters were cloned controlling *gfp* transcrip-tion, which was measured using RT-qPCR. The *pcp* transcript was used as a refer-ence. Individual values and the average ± standard deviation of three biological replicates and two technical replicates are shown. Statistical significance was assessed with a two-sided Student's *t*-test using the mean of technical replicates for each biological replicate. **F** Rationale of microscopy-based pulse chase assay of ribosomes. **G** Decay of labelled ribosomes after the exponential phase in WT (grey) and Δ*ymcA* (green) strains. The mean fluorescence intensity (MFI) per cell was measured by epifluorescence microscopy and normalized to the MFI at timepoint 0. Around 200 cells were analyzed by time point and strain, representing two independent experiments. The boxplot represents the interquartile range (IQR) and the median in the center. Whiskers show the variability outside quartile 1 (Q1) and Q3 and were calculated as Q1-1.5*IQR and Q5 + 1.5*IQR, respectively. The sta-tistical significance was tested using a two-sided Wilcoxon test and adjusted for multiple testing using Holm's method. The half-life (t1/2) was calculated by fitting a one-phase decay function. The data represent the $t_{1/2}$ average ± S.D of two inde-pendent experiments. **H** Ribosome sedimentation profile in 10–50% sucrose gra-dients of WT (grey) and Δ*ymcA* (green) in exponential (OD 0.3), late exponential, and transition phases (as shown in Fig. 2A). The 70S and 100S ribosomes are highlighted with a blue or red box, respectively. The data is representative of three biological replicates.

observed when the r-protein mass fraction is calculated at the transition phase (Fig. 4D).

Given the increased accumulation of ribosomes in the (p)ppGpp[0] Δ*ymcA* strain, we evaluated whether 100S ribosomes could also form in this strain during the transition phase. We observed a strong increase in Hpf levels during the transition phase, including in the (p)ppGpp[0] and (p)ppGpp[0] Δ*ymcA* strains (Supplementary Fig. 8A–B). These elevated Hpf levels are similar in quantity to those of r-proteins, suggesting that they could sustain 100S ribosome formation (Sup-plementary Fig. 8C). Indeed, we observed accumulation of 100S ribosomes in the (p)ppGpp[0] and (p)ppGpp[0] Δ*ymcA* strains (Supple-mentary Fig. 8D), suggesting the presence of a (p)ppGpp-independent pathway of 100S formation.

It was previously observed that (p)ppGpp is required for the transcriptional induction of *hpf* during the transition phase via its σ[H] promoter[45]. However, *hpf* transcription is complex, involving addi-tional predicted promoters for σ[A], σ[B], and σ[E][46–48]. Different, not yet known, signals or specific conditions might be necessary to facilitate the formation of hibernating 100S ribosomes.

The more severe growth defect caused by the absence of (p)ppGpp and the Y-complex, together with the reduced downregulation of ribosome levels in the (p)ppGpp[0] Δ*ymcA* strain (Fig. 4), indicates that both systems are critical for regulating ribosome levels during post-exponential growth.

### Increased ribosome levels lead to protein aggregation during heat shock

Since we identified the Δ*ymcA* strain in a heat-shock competition experiment, we explored whether the Y-complex is involved in low-ering ribosome levels upon such a proteotoxic stress. We evaluated the degradation of 23S and 16S rRNA by the Y-complex before and after heat shock (Fig. 5A, Supplementary Fig. 9A). After 15 min at 50 °C, degradation intermediates are observed in the WT but not in Δ*ymcA*, confirming a role of the Y-complex in ribosome degradation during heat stress (Fig. 5A). Comparing r-protein mass fractions before and during heat shock shows that both WT and Δ*ymcA* downregulate ribosome levels during the experiment. Overall, there is an increase in the levels of r-proteins in the Δ*ymcA* strain already before the heat shock (Fig. 5B), which suggests a role of the Y-complex in maintaining ribosome levels during exponential growth. Notably, the levels and induction of known heat-shock-regulated proteins are not affected in the *ymcA* deletion (Fig. 5C, Supplementary Data 10, Supplementary Fig. 9B). This suggests that chaperone and protease induction during heat stress in the Δ*ymcA* strain is intact.

The downregulation of translation is crucial to limit the increased burden of misfolding and aggregation-prone nascent polypeptide chains on the PQC system. In agreement, a (p)ppGpp[0] strain shows increased aggregate formation during heat shock[7]. We quantified the relative amount of aggregated proteins before and after 15 min of severe heat shock at 54 °C and observed that the Δ*ymcA* strain con-tained significantly higher amounts of aggregated proteins already prior to heat exposure (Fig. 5D). This is consistent with the observed increased ribosome levels of the Δ*ymcA* strain even before heat shock (time point 0 in Fig. 5B). After 15 min of heat shock, Δ*ymcA* shows 20% more aggregated proteins than WT. The (p)ppGpp[0] Δ*ymcA* mutant exhibits even higher protein aggregation than Δ*ymcA* or (p)ppGpp[0] strains alone, further supporting that (p)ppGpp and the Y-complex are critical for the regulation of ribosome levels and translation (Supple-mentary Fig. 9C). Mutants of the Y-complex are also strongly affected by other proteotoxic stresses, such as osmotic and oxidative stress, but not by ethanol (Supplementary Fig. 9D), which further demon-strates the importance of controlling ribosome levels under proteo-toxic stress conditions.

We observed that the Y-complex globally controls the RNase Y-mediated maturation of transcripts and adjusts ribosome levels to the growth state, allowing the endonuclease RNase Y to initiate the processing and degradation of rRNA and r-protein transcripts. The absence of the Y-complex results in an abundance of idle ribosomes, which interfere with the downregulation of translation and increase sensitivity to protein folding stresses.

## Discussion

Bacterial adaptation to environmental challenges depends on their ability to quickly change gene expression. Beyond the well-known regulation of transcription initiation, we provide evidence that post-transcriptional mechanisms, which influence RNA stability or transla-tion efficiency, also contribute to the rapid and tailored cellular response. In this context, the control of RNases plays a critical role. We show that the fitness of *B. subtilis* during heat shock is dependent on certain RNases and the Y-complex, which controls the substrate selection of the quasi-essential endonuclease RNase Y[27] (Fig. 1B–C). Together, our findings demonstrate that the Y-complex – RNase Y system is a regulatory hub that modulates gene expression and the cell's translational capacity during stress response.

Our RNA-seq analyses reveal that the Y-complex may regulate transcripts involved in a wide range of metabolic pathways, develop-mental processes, stress responses, and translation (Fig. 2G, Supple-mentary Fig. 4A–B, Supplementary Data 2, 6–7). RNA processing by the Y-complex-guided RNase Y can create new transcript isoforms[27] or modify regulatory elements such as UTRs[27,36] (Fig. 2F, Supplementary Fig. 3), thereby adjusting gene expression to specific growth condi-tions. This broad regulatory scope of the Y-complex could explain the known pleiotropic phenotypes in these mutants, like decreased spor-ulation, competence, and defective biofilm formation[49–51].

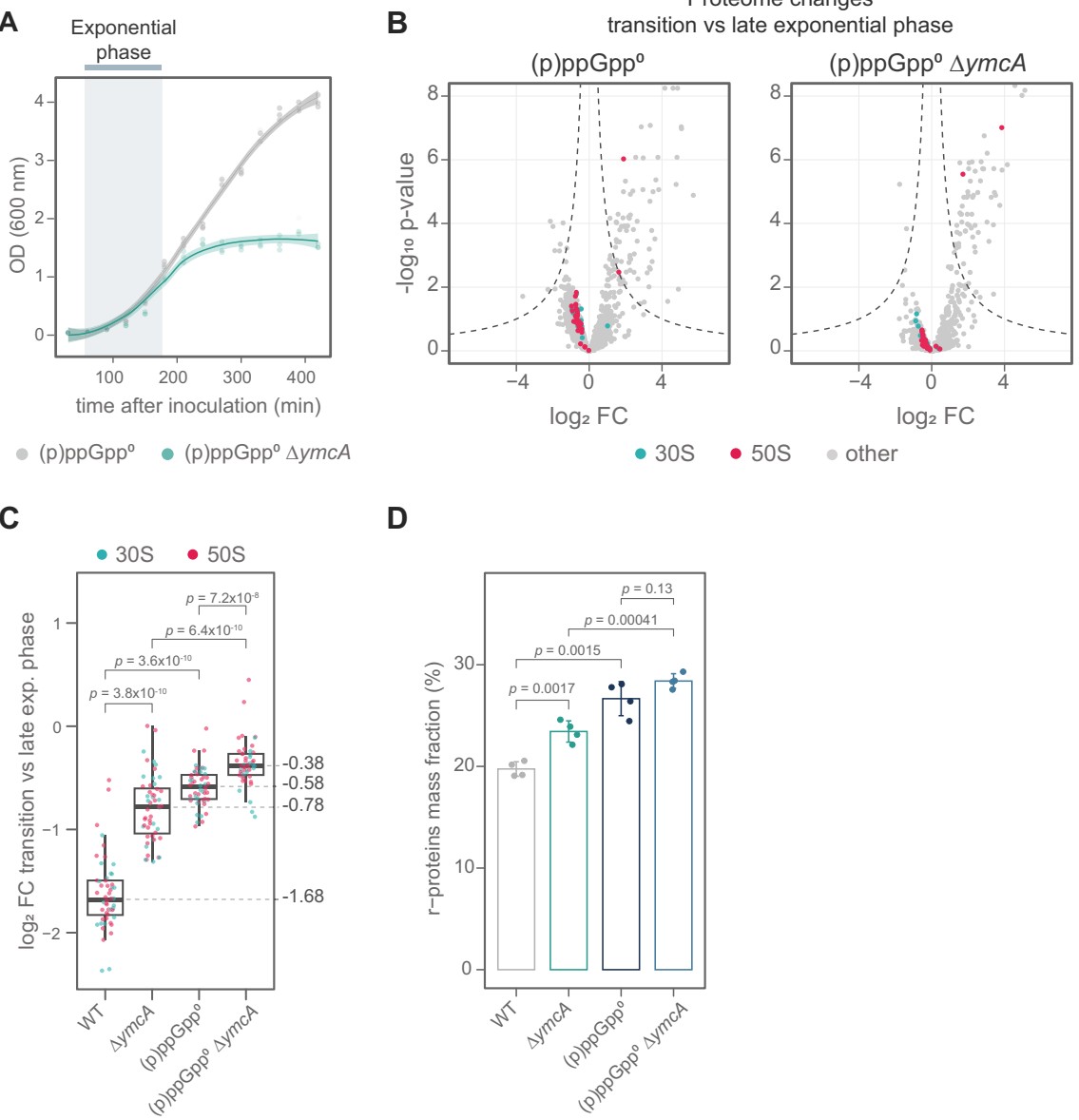

**Fig. 4 | The alarmones (p)ppGpp and the Y-complex are important for controlling ribosome levels. A** Growth curve of (p)ppGpp⁰ and (p)ppGpp⁰ Δ*ymcA* strains in LB at 37 °C. The blue bar marks the exponential phase. Three biological replicates are displayed with a smooth curve regression (LOESS) and a band representing the 95% confidence interval. The growth phases were classified using a semi-log plot. **B** Volcano plot of proteomic changes of (p)ppGpp⁰ and (p)ppGpp⁰ Δ*ymcA* cultures comparing transition to exponential phase. The proteins composing the small (30S) and major (50S) ribosome subunits are highlighted in cyan and magenta, respectively. The data shows the average of four biological replicates. The *p*-values were calculated using a two-sided Student's *t*-test and adjusted using the Benjamini-Hochberg method. **C** Comparison of r-proteins downregulation in WT, Δ*ymcA*, (p)ppGpp⁰, and (p)ppGpp⁰ Δ*ymcA* during the transition phase

compared to the late exponential phase. The boxplot represents the interquartile range (IQR) and the median in the center. Whiskers show the variability outside quartile 1 (Q1) and Q3 and were calculated as Q1-1.5*IQR and Q5 + 1.5*IQR, respectively. Proteins are highlighted like in **B**. Each dot represents the average of four biological replicates. Alternative r-proteins were excluded from the analysis. The statistical significance was tested using a two-sided paired Wilcoxon test and adjusted for multiple testing using Holm's method. **D** Proteome fraction (in %) of r-proteins mass, measured by mass spectrometry. The r-protein mass fraction was calculated for the indicated strains. Individual values and the average ± standard deviation of four biological replicates are shown. The statistical significance was tested using a two-sided Student's *t*-test and adjusted for multiple testing using Holm's method.

Among these diverse functions in RNA processing, we discovered that the Y-complex initiates ribosome degradation at the end of exponential growth by directing RNase Y to cleave 23S and 16S rRNA (Fig. 3A–B, Supplementary Fig. 7A–D), thereby decreasing the ribosome levels. This finding is consistent with previous suggestions that RNase Y can degrade rRNA in aging *B. subtilis* spores[52].

The exact nature of how the Y-complex (YmcA, YlbF, and YaaT) associates with RNase Y is not yet known. Recent evidence indicates

that YmcA and YlbF transfer YaaT to RNase Y, forming a stable, membrane-localized YaaT-RNase Y complex[40]. However, the deletion of any of the Y-complex members leads to a strong decrease in the processing of target mRNA[27], indicating that all Y-complex proteins are equally indispensable for RNA processing. Based on the dynamic localization of YmcA and YlbF between the cell membrane and the cytosol[53], we propose that these Y-proteins may also recognize substrates, including abundant ribosomes, and direct them to RNase Y for cleavage.

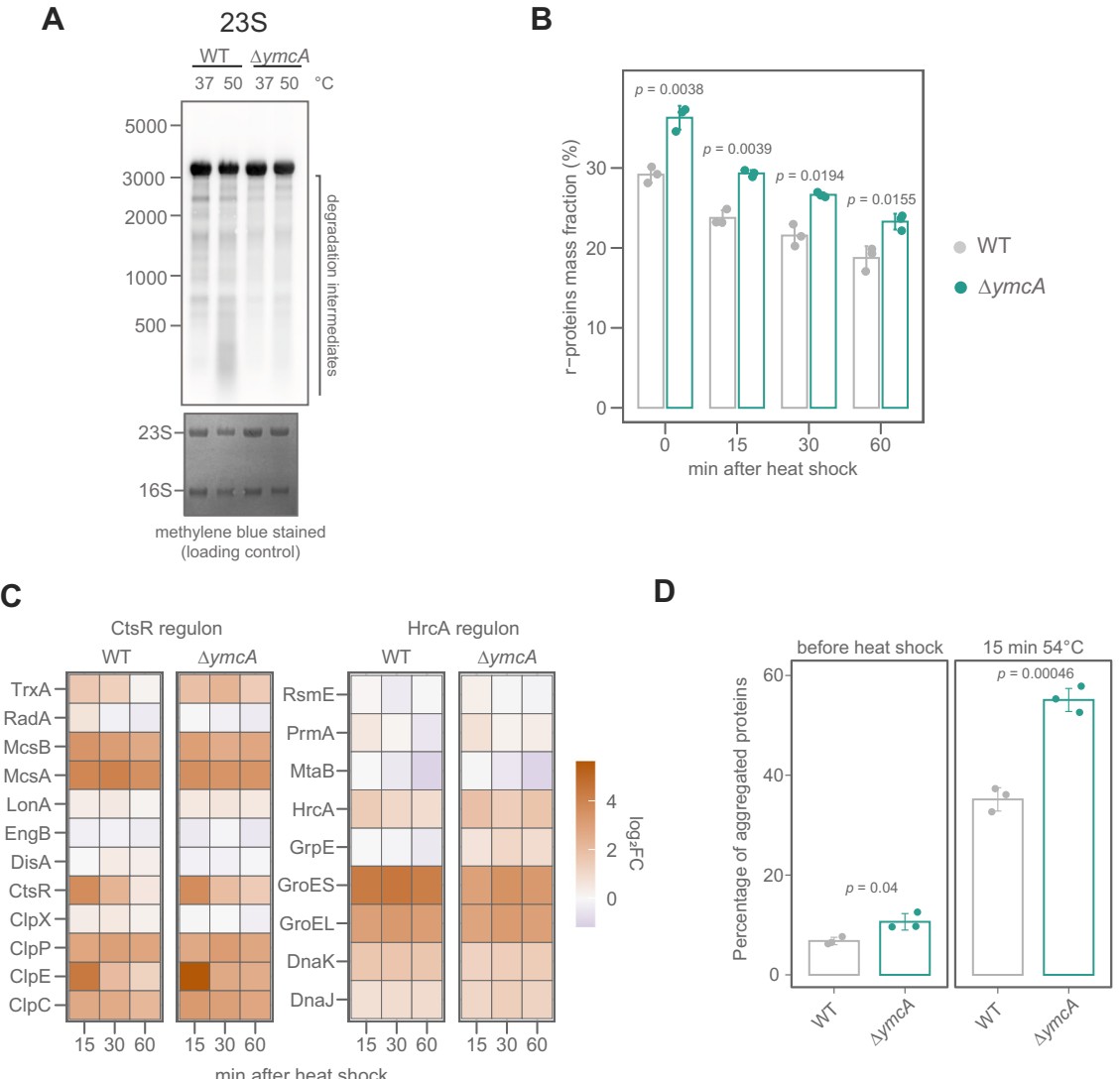

**Fig. 5 | Control of ribosome levels and protein homeostasis in *ymcA* mutant during heat shock. A** Northern blot against 23S of WT and Δ*ymcA* cells at exponential phase before or after a heat shock. The methylene blue-stained membrane is shown as a loading control. This image is representative of three biological replicates. **B** Proteome fraction (in %) of r-proteins mass, measured by mass spectrometry. Individual values and the average ± standard deviation of three biological replicates are shown. The statistical significance was tested using a two-sided Student's *t*-test and adjusted for multiple testing using Holm's method. **C** Induction of CtsR and HrcA regulons after heat shock (50 °C) in WT and Δ*ymcA*. Proteins were measured by mass spectrometry. The color bar shows the average log₂ fold change compared to the control before heat shock (three biological replicates). **D** Percentage of aggregated proteins before and after a 15 min heat shock at 54 °C in WT (grey) and Δ*ymcA* (green). Individual values and the average ± standard deviation of three biological replicates are shown. The statistical difference was tested using a two-sided Student's *t*-test.

The two [4Fe-4S]$^{+2}$ clusters in the Y-complex[54] could act as sensors for cellular growth, energy, and stress, thereby affecting how the complex controls RNase Y. This regulatory complex is localized at the cell membrane, where RNase Y forms a dynamic degradosome-like network with other RNases, helicases, and glycolytic enzymes[55–57]. This spatial co-localization may enable a sequential degradation process: the Y-complex-guided RNase Y performs initial cleavage on targets like rRNA, making fragments susceptible to further degradation by exonucleases such as PNPase, and RNase J as observed with YaaT pull-down experiments (Supplementary Fig. 7B). Here, we propose that the Y-complex targets ribosomes to RNase Y and probably other members of the membrane-localized degradosome-like network (Fig. 6). This suggests complex spatial control of RNA processing and ribosome decay, separate from transcription and translation in the cytoplasm.

Most of the cleavages we observed in rRNA were in the 23S (Fig. 3A); however, the Y-complex and RNase Y can also initiate 16S

degradation in the late exponential phase and during heat shock (Fig. 3A, Supplementary Fig. 7A, Supplementary Fig. 9A). Recently, it was observed that the exoribonuclease RNase R alone can degrade 16S rRNA[58], suggesting that endonucleolytic cleavages might not be strictly necessary for 16S degradation. RNase R is induced as part of the general stress response[59], suggesting a possible stress-specific and Y-complex-independent degradation pathway for 16S rRNA.

Complete ribosome decay requires the concurrent degradation of r-proteins (Fig. 6). These proteins might be passively released from the ribosome into the cytoplasm during the degradation of the rRNA, which would also change their interaction with rRNA. Therefore, these released r-proteins might misfold and aggregate, and can subsequently be degraded by AAA+ protease complexes. Alternatively, these AAA+ protease complexes could be actively recruited to the site of ribosome decay to simultaneously extract and degrade the r-proteins along with rRNA degradation. Thereby, AAA+ protease complexes

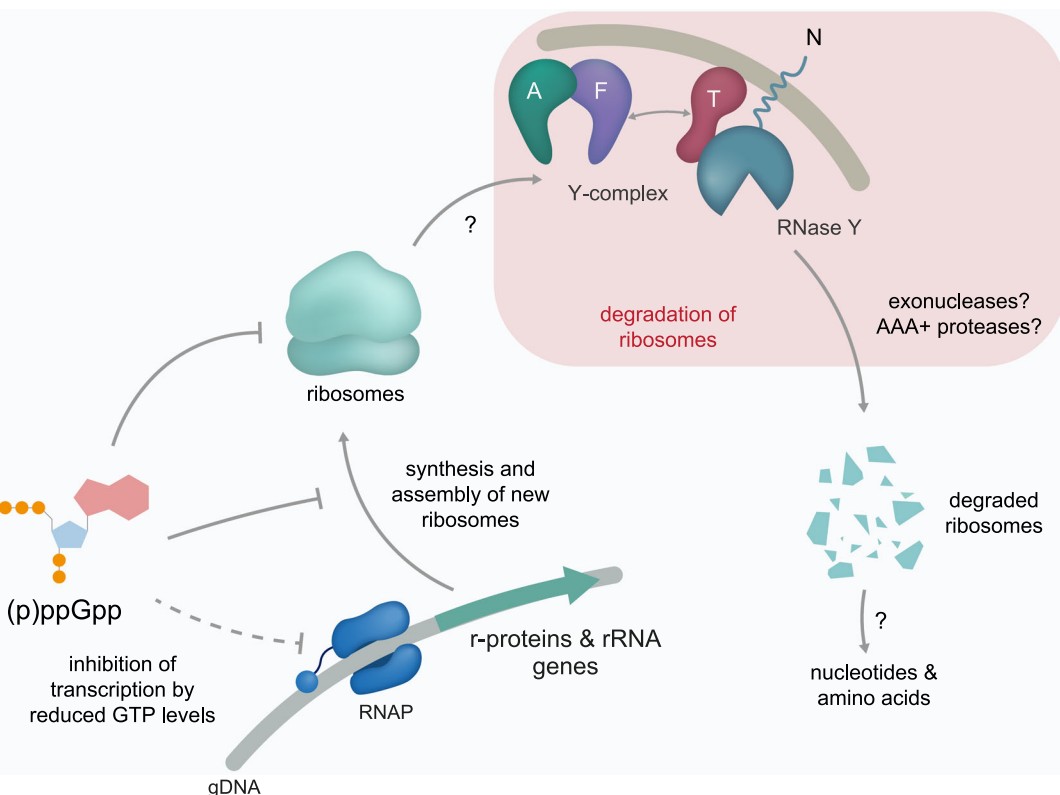

**Fig. 6 | Control of ribosome levels by (p)ppGpp and the Y-complex.** The alarmone (p)ppGpp reduces ribosome biogenesis by indirectly inhibiting the transcription of rRNA and r-protein mRNAs. This indirect inhibition occurs in promoters containing a G as the initiating nucleotide (+1), being then sensitive to changes in GTP concentration in the cell[18]. This connection is shown with a dotted line as GTP levels have been shown to drop after (p)ppGpp biosynthesis during amino acid starvation or other starvation-inducing conditions, but recent reports show mild or no decrease in GTP levels at the transition phase[99]. (p)ppGpp also inhibits GTPases that are involved in new ribosome assembly. Additionally, (p)

ppGpp reduces translation by binding to specific GTPases. The Y-complex can initiate the degradation of ribosomes, which need to be transported to the membrane by unknown mechanisms. Exonucleases and AAA+ protease complexes might assist in ribosome degradation for the complete degradation of rRNA and r-proteins, which could partially replenish the nucleotide and amino acid pools during starvation. Some of the elements in this figure are adapted from illustrations in the Driller et al.[21] review, which was published under a CC-BY 4.0 license (https://creativecommons.org/licenses/by/4.0/).

such as ClpCP with its adaptor protein and kinase McsB or the membrane-associated FtsH, which maintain protein homeostasis in *B. subtilis*[60], could be involved in the proteolysis of these r-proteins (Supplementary Fig. 7B).

Strikingly, the Y-complex additionally processes transcripts encoding r-proteins and other translation factors (Fig. 2G), such as BipA, EF-Tu, and EF-G (Supplementary Fig. 3E–F), potentially regulating their translation or leading to their decay. This is supported by the findings that the Y-complex is necessary to process some r-protein transcripts at the mid-logarithmic phase[27], and RNase Y depletion in this growth phase leads to an increased abundance of r-protein transcripts, such as *rplS, rpsO, rpmA, rpsD, rpsB*, and *rplU*[61].

The control of translation is an important component of many stress response mechanisms. During nutrient starvation, down-regulating translation allows the reallocation of resources to biosynthetic pathways[62]. During proteotoxic stresses, it could relieve the overwhelmed PQC system by reducing the synthesis of new proteins that could potentially aggregate[21]. The stress transcription factor Spx and the alarmones (p)ppGpp regulate ribosome levels and activities in *B. subtilis*, and the lack of both impairs growth at 50 °C[7]. Here, we observe a link between the regulation of ribosome levels through synthesis and decay. The severe growth defect of cells lacking both (p)

ppGpp and the Y-complex indicates that both of these systems are important for transitioning from exponential to stationary phase. This results in elevated ribosome levels and increased accumulation of protein aggregates (Supplementary Fig. 9C). Interestingly, (p)ppGpp has been shown to directly modulate the activity of RNases in other species, such as the Nudix hydrolase RppH in *E. coli* and PNPase in *Streptomyces*[63,64], linking (p)ppGpp to RNA metabolism. Thereby, the alarmones can be involved in rRNA degradation via the Y-complex and RNase Y, or another RNase.

In stress conditions, the collective activities of (p)ppGpp and the Y-complex in regulating ribosome levels by inhibiting their synthesis and initiating their degradation enable resource allocation to other biosynthetic pathways (Fig. 6).

These strategies are not exclusive to *B. subtilis*, as different organisms use various mechanisms to degrade ribosomes. These must specifically recognize the ribosome or its subunits to disassemble them and target rRNA and r-proteins for degradation. For example, *E. coli* cells transitioning into the stationary phase also initiate ribosome degradation by the membrane-associated endonuclease RNase E[65,66], which is functionally equivalent to RNase Y in *B. subtilis*[67]. RNase E forms the scaffold of the membrane-associated degradosome network in *E. coli*[68]. This could allow for the combined action of endo- and

exonucleases, facilitating the complete degradation of the ribosome[67,69]. In addition, it was observed that *E. coli* r-proteins are degraded by the Lon AAA+ protease during starvation[70]. In eukaryotic cells, nutrient starvation also triggers ribosome degradation via the ubiquitin-proteasome system or via specialized autophagy pathways, such as ribophagy[71]. Recent studies have described how stress- or starvation-induced degradation of the 40S ribosome involves the ubiquitylation of proteins S3 and S5. This allows RIOK3 binding, leading to the decay of 18S rRNA by unknown RNases[72–74].

In summary, we identified the Y-complex as a key regulator of ribosome abundance in *B. subtilis*. The Y-complex regulates the translational activity by triggering ribosome decay and processing transcripts for new ribosomal components. Our findings contribute to the understanding of ribosome degradation by identifying a direct pathway and illustrating how this process is integrated with other stress response mechanisms to control cellular growth and survival. Further studies are required to understand the molecular signals that trigger ribosome degradation and how the decay of rRNA and r-proteins is temporarily and spatially coordinated.

## Methods

### Growth conditions

Unless stated the contrary, strains (Supplementary Data 11) were streaked on LB plates (5 g/l yeast extract, 10 g/l tryptone agar, 10 g/l NaCl, 1.5% agar) containing the required antibiotics for selection (1 μg/ml erythromycin + 25 μg/ml lincomycin, 150 μg/ml spectinomycin or 10 μg/ml kanamycin) and incubated overnight at 37 °C. A single colony was inoculated in 3 ml of LB medium and grown overnight at 37 °C and 180 rpm, then used as the inoculum for subsequent experiments.

### Strain construction

For constructing the strains Δ*ymcA::erm*[R] and Δ*yaaT::erm*[R], the loci containing the resistance cassette were PCR-amplified from the respective strains of the BKE library[20] and transformed in *B. subtilis* WT. The deletions of Δ*rel::kan*[R] and Δ*ylbF::erm*[R] were constructed as described in Ref. 20. For promoter activity reporter strains, the *gfp*(mut2) gene was cloned downstream of the promoters of interest, using the integrative plasmid pDR111 as a backbone. For the inducible RplA(uL1)-HaloTag reporter, the native *rplA* gene was cloned under the control of the IPTG-inducible *hyperspank* promoter and fused in frame to a codon-optimized HaloTag sequence, using the integrative plasmid pDR111 as the backbone.

For transformation, 600 μl of an overnight culture of the receptor strains were diluted in 5 ml competence medium (17.5 g/l K$_2$HPO$_4$, 7.5 g/l KH$_2$PO$_4$, 2.5 g/l (NH$_4$)$_2$SO$_4$, 1.25 g/l tri-sodium citrate x2 H$_2$O, 0.5% glucose, 7 mM MgSO$_4$, 0.1 mg/ml L-tryptophan, 0.02% casamino acids, 0.22 g/ml ammonium iron citrate) and grown at 37 °C, 180 rpm, for 3 h. Subsequently, 5 ml of starvation medium (17.5 g/l K$_2$HPO$_4$, 7.5 g/l KH$_2$PO$_4$, 2.5 g/l (NH$_4$)$_2$SO$_4$, 1.25 g/l tri-sodium citrate x2 H$_2$O, 0.5% glucose, 7 mM MgSO$_4$) were added and grown for 2 more hours. One ml of this culture was incubated with the DNA to be transformed and grown for 2 h at 37 °C and 180 rpm. Positive clones were selected on LB agar containing the required antibiotics. Genetic modifications were verified using PCR and DNA sequencing.

### Genetic screening during heat shock

The BKE library[20] was replicated on LB agar plates containing 1 μg/ml erythromycin. Liquid cultures were prepared in 200 μl LB medium supplemented with 1 μg/ml erythromycin and grown for 42 h at 30 °C 200 rpm. All mutant strains were pooled and mixed for 15 min using a magnetic stirrer. The pooled culture was centrifuged at 6240 x *g* for 10 min at 18 °C. The pellet was resuspended in LB medium to an OD$_{600}$ of 65 and mixed with an equal volume of 50% glycerol. The pooled library was aliquoted and kept at −80 °C.

For each replicate, a tube of the pooled library was quickly unfrozen in a water bath at 30 °C and used to inoculate 75 ml LB medium in 500 ml flasks to a starting of OD$_{600}$ 0.05. The culture was grown in a water bath at 30 °C, 180 rpm, until OD$_{600}$ 0.4. Fifty ml of culture were split into two 250 ml flasks. One flask was transferred to 50 °C, 180 rpm for heat shock. The other flask was used as a control sample and was kept at 30 °C. One ml samples were collected for gDNA purification at 0 and 2 h post temperature shift. Samples were pelleted for 2 min at 15,870 x *g* and washed once with 500 μl PBS. gDNA was purified using the Wizard® Genomic DNA Purification Kit (Promega, A1120) and stored at −20 °C until further processing.

The strain-specific barcodes were amplified using indexed primers (Supplementary Data 12) with the Collibri Library Amplification Master Mix (ThermoFisher, A38539050) according to the manufacturer's instructions, with 100 ng of gDNA as input, for 22 cycles. Barcodes were gel-purified, quantified, pooled equimolarly, and sequenced using the Illumina NovaSeq2 single-read method at the Sequencing Core Facility of the Max Planck Institute for Molecular Genetics (Berlin, Germany).

To calculate the fitness score of each deletion mutant from barcode abundance, BEAN-counter v2.6.1[75] was used with the following settings. A fixed amplicon read structure was used with an index at position 0, the common primer sequence GCAGGCGAGAAAGGAGAG at position 9, and the mutant barcode at position 27 of read 1. All other settings were used with default values. The sample before heat shock served as a reference to which all other time points were compared. All comparisons were performed with three biological replicates.

Fitness data was further analyzed using the R statistical computing language (v4.3.3). For annotation, protein-related information was fetched using the Uniprot REST API (29 February 2024). Average fitness per gene, condition, and time point was calculated as the arithmetic mean of the fitness score from three biological replicates. Differential fitness Δ*F*, e.g. for comparison of treatment *versus* control at a particular time point, was calculated as $\Delta F = F_{treatment} - F_{control}$. A two-sided Student's t-test was performed to evaluate statistical significance for each gene, and the obtained *p*-value was corrected using the Benjamini-Hochberg procedure[76].

### Growth curves

Twenty-five ml of LB medium were inoculated with an overnight culture to a starting OD$_{600}$ 0.05 in 250 ml Erlenmeyer flasks. Cultures were grown at 37 °C 180 rpm in a water bath. OD$_{600}$ was monitored every 30 min. The average and standard deviation of three biological replicates were calculated.

### Heat shock survival

Cultures were inoculated in 25 ml LB medium to OD$_{600}$ 0.05 and incubated in a water bath at 37 °C 180 rpm. At OD$_{600}$ 0.3 – 0.4, the cultures were exposed to a heat shock at 54 °C for 2 h, and samples for viability measurement were obtained every 30 min. Viability was measured by spotting 4 μl of 1:10 serial dilutions in PBS of each sample on LB agar. The plates were incubated overnight at 30 °C, and the colony-forming units (CFU) were counted. The CFU/ml was calculated for each sample, and relative viability was determined using samples before heat shock as a reference.

### Isolation of protein aggregates

Twenty-five ml of culture were harvested before and 15 min after heat shock at 54 °C by centrifugation for 5 min at 11,620 x *g*. The pellets were washed once with buffer A (100 mM HEPES pH 8.0, 150 mM NaCl) and lysed by bead-beating in buffer B (100 mM HEPES pH 8.0, 150 mM NaCl, cOmplete EDTA-free protease inhibitor) using 0.1 mm glass beads. Afterwards, samples were centrifuged at 1000 x *g* for 2 min to obtain the cell extract. The soluble fraction was separated from protein aggregates by centrifugation for 15 min at 11,000 x *g* at 4 °C. To remove

membrane proteins, the aggregates were resuspended in buffer B containing 1% Triton X-100 and incubated for 1 h at 4 °C. Protein aggregates were recovered by centrifugation at 11,000 x $g$, 4 °C for 15 min and resuspended and incubated as described above. Protein aggregates were washed again with buffer B + 0.5% Triton X-100, and the detergent was removed in a final wash using buffer B. Pellets containing protein aggregates were resuspended in buffer B + 2% SDS by boiling for 5 min at 95 °C. Protein concentration was measured using the Rapid Gold BCA Protein-Assay-Kit (Pierce™, A53225). The percentage of aggregated protein was calculated to the total protein content in the cell extract.

## Spot test
Overnight cultures were diluted 1:1000 in LB medium without antibiotics and incubated at 37 °C 180 rpm for 7–8 h. The cultures were diluted to $OD_{600}$ 1 in fresh LB medium in a 96-well plate. Seven serial 1:10 dilutions were prepared in 1x PBS. Four µl of each dilution were spotted on LB agar containing different stressors (concentration indicated in the figures). Plates were freshly prepared before use.

## RNA purification
Samples for RNA purification were collected at the indicated time points: exponential phase ($OD_{600}$ ~ 0.3), late exponential phase ($OD_{600}$ ~ 1) and transition phase (100 min after late exponential phase). Rifampicin was added to a final concentration of 100 µg/ml for 10 min.

Samples were harvested by mixing with the same volume of cold 1:1 acetone:ethanol and centrifugated at 9,410 x $g$ for 5 min, 4 °C. RNA purification was performed according to ref. 32 with slight modifications: The pellet was washed twice in 1 ml TE-sucrose buffer (50 mM Tris-HCl pH 8.0, 10 mM EDTA, 25% sucrose) and centrifugated at 9390 x $g$ for 5 min, 4 °C. Cells were resuspended in 130 µl lysis buffer (20 mM Tris-HCl pH 8.0, 50 mM EDTA, 20 % sucrose) supplemented with 3.2 mg/ml lysozyme and incubated on ice for 5 min. Two hundred µl of lysis executioner (2% SDS, 1 mg/ml Proteinase K) were added to the samples and incubated at 95 °C for 1.5 min. Samples were mixed with 1 ml of TRIzol (Ambion) and incubated at room temperature. After 5 min, 250 µl chloroform was added to each sample and mixed by vortexing. After 10 min at room temperature, the samples were centrifuged for 10 min at 11,360 x $g$. The RNA was isopropanol-precipitated at −20 °C overnight. RNA was pelleted at 11,360 x $g$ for 20 min at 4 °C, washed with 1 ml 70% ethanol followed by another centrifugation at 15,870 x $g$ for 10 min at 4 °C. The pellet was air-dried and resuspended in 50 µl nuclease-free water.

## ISCP RNA sequencing
RNA sequencing was performed as previously described[77]. In brief, total RNA was treated with TURBO DNase (Ambion, AM2238), depleted of ribosomal RNA using the riboPOOL rRNA depletion kit for *B. subtilis* (siTOOLS; two reactions with 4.5 µg each) and purified using the RNA Clean & Concentrator-5 kit (RCC-5; Zymo Research). RNA 5′ triphosphates were converted to monophosphates using RppH (NEB), and RNA was recovered using phenol:chloroform:isoamyl alcohol extraction followed by ethanol precipitation. RNA ends were repaired using T4 Polynucleotide Kinase (Thermo Scientific), before RNA was purified using the RCC-5 kit and fragmented using an M220 Focused-ultrasonicator (Covaris) as described by the manufacturer but with a treatment time of 100 s (microTUBE AFA Fiber Snap-Cap). After purification using the RCC-5 kit, sequencing libraries were prepared from the recovered RNA using the NEXTflex Small RNA-Seq Kit v3 (Bioo Scientific) according to the manufacturer's protocol (including step G). Libraries were purified as described in step I of the NEXTflex Rapid Directional qRNA-Seq Kit (Bioo Scientific) using Agencourt AMPure XP Beads (Beckman Coulter; 0.9x bead-to-sample ratio). Sequencing was performed on an Illumina NovaSeq 6000 in paired-end mode with a read length of 100 bp at the Sequencing Core Facility of the Max Planck Institute for Molecular Genetics (Berlin, Germany).

## Processing of RNA sequencing data
FastQC (v0.11.5) was used to assess the quality of the data. Reads were filtered with a minimum quality score of 10 and a length of at least 18 nt, cleaned from adapter sequences using Cutadapt (v3.5)[78], and mapped against the *B. subtilis* reference genome (NC_000964.3) using STAR (v2.7.10a_alpha_220314)[79] in 'random best' and 'end to end' modes. Resulting BAM files were sorted and indexed with Samtools (v1.9)[80], and PCR duplication artefacts were removed using UMI Tools (v1.1.0)[81]. Gene counts of annotated RefSeq genes were determined with featureCounts (v2.0.3)[82] and differentially expressed genes (DEG) were calculated with three replicates per condition and time point using DESeq2 (v1.24.0)[83]. $p$-values were corrected for multiple testing using the Benjamini-Hochberg method[76]. To call DEGs, a threshold of adjusted $p$-value < 0.05 and absolute log2 fold change > 1 was used. Genome coverage profiles (total, 5′ and 3′ ends) were generated using a custom script utilizing the HTSeq library (v2.0.2)[84]. We performed the aforementioned analysis steps using a customized pipeline implemented in Snakemake (v7.14.0)[85].

## Identification of Y complex cleavage sites
The Y-complex-dependent cleavage positions were identified following the previously published procedure[77]. In brief, the genome coverage data was prefiltered with a count per million (cpm) value ≥ 0.05, and only the RNA ends displaying a cpm ≥ 5 were further analyzed. We carried out differential expression analysis of normalized 5′ and 3′ read ends by comparing WT vs. Δ*ymcA* at the late exponential and transition phase. Y-complex-dependent ends were identified using edgeR (v3.28.0)[86] with absolute log2 fold change ≥ 1 and FDR < 0.05, and kept only if present in both comparisons. These results were further filtered with additional parameters, i.e., the "proportion of ends (cleavage ratio)" and the "ratio of WT and Δ*ymcA* proportion of ends" as was previously described[77]. Moreover, to reduce the potential set of false positive processing sites even further, we applied an additional cut-off on the "proportion of ends" parameter, i.e., we kept only processing sites with a cleavage ratio of the reference strain (WT) ≥ 95%-percentile of the cleavage ratio of the mutant strain.

Given the high sequence conservation of the different rRNA copies, we performed additional analyses to minimize the effect of multimapping reads. We performed the mapping using each rRNA copy independently to determine RNA ends depending on *ymcA*. In parallel, we used the consensus rRNA sequence to map ends and selected the Y-complex-dependent ends detected by the three independent approaches.

## Impact of processing sites on transcript and operon expression
To examine the impact of cleavage sites on transcript and operon expression, as well as other genomic features such as untranslated regions (UTRs) and regulatory elements, we calculated transcript per million (TPM) values from variance-stabilized data for each annotated RefSeq gene (transcript level) and for each operon as defined in the BSGatlas (v1.1)[46] using the DESeq2 package and a custom Python script. Furthermore, all Y-complex-dependent cleavage sites were mapped to various genomic features, including coding regions, UTRs, terminator regions, and riboswitches, as defined in the BSGatlas[46] (Supplementary Data 2).

## Processing site sequence logo
The sequence logos were generated with Logomaker (v0.8)[87], specifically for 5′ and 3′ end U-type cleavage sites. For this, we extracted 30 nt sequences centered at the cleavage site and calculated the logos with a GC content of 43.5%.

## RNA secondary structure prediction

The RNA structure surrounding Y-complex-dependent cleavage sites was estimated by calculating the MFE ($\Delta G$ in kcal/mol) utilizing RNA-fold (v2.6.4)[88]. This analysis employed a sliding window approach with 50 nt sequences, focusing on a 200 nt region centered around the position of interest. The average MFE at each nucleotide was then calculated. To calculate and visualize the background folding energy, we randomly draw potential processing sites from the genome determined with the corresponding 5′ or 3′ end motif and calculate the average MFE at each nucleotide as described above. Potential processing sites were drawn 50 times independently.

## Overrepresentation analysis

Annotations of operons were obtained from BSGatlas[46] and gene categories from Subtiwiki (24 July 2023)[48]. The overrepresentation analysis was performed using the R statistical computing language (v4.2.2) and the Bioconductor package Category (v.2.64.0).

## Northern blot

Biotinylated RNA probes were prepared using the HighYield T7 Biotin11 RNA Labeling Kit (UTP-based) (Jena Bioscience, RNT-101-BIOX) from PCR amplified fragments containing the promoter for the T7 RNA polymerase (Supplementary Data 12). Probes were purified using the Monarch® Spin RNA Cleanup Kit (NEB, T2040) according to the manufacturer's instructions. Probes were designed taking into consideration the predicted cleavage sites.

One and a half µg of total RNA were separated on a denaturing agarose gel (1% agarose, 20 mM MOPS, 5 mM NaOAc, 1 mM EDTA, 6.6% formaldehyde, pH 7.0) in 1x AGE-NB Running Buffer (20 mM MOPS, 5 mM NaOAc, 1 mM EDTA, 0.74% formaldehyde, pH 7.0). The gel was washed 3 times with milli-Q water and once with 10x SSC buffer, and the nitrocellulose membrane was washed once with milli-Q water and once with 10x SSC. The RNA was transferred using a vacuum blotter at 300 mbar covered with 10x SSC for 1.5 h. After transfer, the membrane was washed for 2 min with 6x SSC, once with water, and dried and crosslinked at 254 nm 0.12 J twice. The membrane was developed with methylene blue, and the staining was removed by washing with 0.2x SSC containing 1% SDS, followed by washing with milli-Q water. The hybridization of 16S or 23S rRNA probes was performed using the North2South Chemiluminescent Hybridization and Detection Kit (Thermo) and North2South Hybridization Stringency Wash Buffer (2X) according to the manufacturer's instructions. Blots were developed in a Vilber imaging system.

## RT-qPCR

Twenty µg of total RNA were treated with TURBO DNase (Ambion, AM2238) and purified using the Monarch® Spin RNA Cleanup Kit (NEB, T2040). Subsequently, cDNA was prepared using the LunaScript RT Mastermix (NEB, E3025) and 2.5 µg of DNase-treated RNA as input. Quantitative PCR was performed using the Luna Universal qPCR Master Mix (NEB, M3003) in a QuantStudio 5 Real-Time PCR System (Applied Biosystems). To calculate changes in *gfp* transcription, samples from the late exponential and transition phases were compared to those from the exponential phase, with *pcp* serving as a reference gene using the Pfaffl method[89]. Primers used for RT-qPCR are listed in Supplementary Data 12.

## Ribosome sedimentation in sucrose gradients

Cultures of WT and Δ*ymcA* were grown until the exponential phase ($OD_{600}$ ~ 0.3), late exponential ($OD_{600}$ ~ 1), and transition phase (100 min after the late exponential phase). Chloramphenicol was added at a concentration of 100 µg/ml 2 min prior to sample collection. Fifty ml of culture were pelleted at 4,420 x $g$ for 10 min at room temperature. Pellets were transferred to ice and resuspended in 2 ml of lysis buffer (20 mM Tris-HCl pH 8.0, 100 mM $NH_4Cl$, 10 mM $MgCl_2$, 0.4% Triton X-100, 0.4% NP-40, 1 mM chloramphenicol, 100 U/µl DNase I), and flash-frozen. Cells were lysed under cryogenic conditions using 5 cycles of mix-milling at 30 Hz for 1 min. Lysates (240 µg RNA) were laid over a 10–50% sucrose gradient in 20 mM Tris-HCl pH 8.0, 100 mM $NH_4Cl$, 10 mM $MgCl_2$, 0.5 mM DTT and resolved at 217,870 x $g$ for 3 h at 4 °C. Gradients were analyzed using a piston fractionator (Biocomp instruments), and absorbance at 260 and 280 nm was monitored.

## *rplA*(uL1)-HaloTag pulse-chase experiment

Cultures were inoculated into LB medium at an $OD_{600}$ of 0.05 and incubated in a water bath at 37 °C, 180 rpm. Two flasks were inoculated for each strain: one for the pulse-chase experiment and another for producing conditioned media for the chase. For the pulse-chase experiment, cultures were grown to an $OD_{600}$ of 0.45–0.5. RplA (uL1)-HaloTag expression was induced by adding IPTG to a final concentration of 0.5 mM, together with the HaloTag TMR ligand (Promega, G825A) to a final concentration of 5 nM. After 20 min of incubation in the dark at 37 °C, 180 rpm, bacterial cells were harvested at 1,500 x $g$ for 5 min at room temperature. In parallel, the second culture was filtered through a sterile 0.22 µm membrane, the conditioned media was recovered, and 7-Bromo-1-heptanol was added to a final concentration of 10 µM (chase media).

TMR-labelled bacterial pellets were washed, resuspended in 1 mL of warm chase media, and transferred to a fresh culture flask with 24 mL of the same media to start the chase. Cultures were further incubated at 37 °C, 180 rpm in the dark. Samples for microscopy were collected at different chase time points (0, 30, 60, and 90 min). For this, 1 mL of culture was centrifuged for 2 min at 1,500 x $g$, room temperature, followed by washing with filtered 1x PBS. Cell pellets were resuspended in 100 µL filtered 1x PBS and stained by adding 1 µl of 1 mg/mL DAPI. After incubation for 1 min at 37 °C, cells were harvested as described above and resuspended in filtered 1× PBS. One microliter of this suspension was spotted onto agar pads (1.5% low-melting-point agarose in filtered 1× PBS). Images were acquired with 50 ms exposure time for phase contrast, 50 ms for the DAPI stain (Blue channel), and 800 ms for visualization of TMR-labelled HaloTag (TX-Red channel). Micrographs were processed in ImageJ using MicrobeJ[90] for cell segmentation and fluorescence intensity measurements. Around 100 cells were evaluated per strain, time point, and replicate. Mean fluorescence intensity (MFI) per cell was normalized to the start of the chase, and the half-life ($t_{1/2}$) was calculated by fitting the MFI at each time point into a one-phase decay function.

## Pulldown of YaaT

Cultures of in-locus C-terminal FLAG-tagged *yaaT* strain were grown until the late exponential phase (OD600 ~ 1) and performed in three biological replicates. Two hundred ml of culture were pelleted at 24,420 x $g$ for 2 min. Pellets were transferred to ice and resuspended in 1 ml of resuspension buffer (20 mM HEPES-KOH pH 7.5, 30 mM $NH_4Cl$, 6 mM Mg-Acetate, 0.5 mM TCEP) supplemented with cOmplete EDTA-free protease inhibitor), and flash-frozen. Cells were lysed under cryogenic conditions using 3 cycles of mix-milling at 30 Hz for 1 min. Lysates were stored at −80 °C until further processing. Lysates were quickly melted at 30 °C, incubated for 10 min on ice with RNase-free DNase I (Roche), and clarified by centrifugation at 16,730 x $g$ for 5 min at 4 °C. DDM (n-dodecyl β-D-maltoside) was added to a 200 µM final concentration. The lysate (input sample) was incubated with 200 µL of anti-FLAG M2 affinity gel (A2220, Sigma) for 2 h at 4 °C with constant shaking. Beads were washed 5 times with resuspension buffer supplemented with 200 µM DDM. Proteins were eluted for 45 min in resuspension buffer supplemented with 200 µM DDM and 150 µg/ml of 3x FLAG peptide (Sigma) at 4 °C with constant shaking. Proteins were identified and quantified using DIA-MS.

## Sample preparation for MS-based proteomics

Samples were collected at the indicated time points: late exponential phase ($OD_{600}$ ~ 1) and transition phase (100 min after the late exponential phase) in four biological replicates. For the proteomic changes during heat shock, cultures were grown at 37 °C until $OD_{600}$ ~ 0.3 and shifted to 54 °C. Samples were collected at 0, 15, 30, and 60 min after heat shock, and it was performed in three biological replicates. Ten ml of culture were collected by centrifugation at 4,420 x $g$ for 10 min, and the pellet was stored at −80 °C until further processing.

Cell pellets were resuspended in 1 ml of cold PBS and centrifuged at 15,870 x $g$ for 5 min at 4 °C. The pellet was washed once more with PBS, resuspended in 150 µL of lysis buffer (100 mM HEPES pH 8.0, 2% SDS, 20 mM TCEP, 60 mM CAA), and lysed using bead beating. Cell debris was discarded by centrifuging at 15,870 x $g$ for 5 min at 4 °C. The supernatant was boiled at 95 °C for 5 min, cooled down to room temperature, and the protein concentration was estimated using the Rapid Gold BCA protein assay kit (Pierce). Samples were flash-frozen in liquid nitrogen and stored at −80 °C until further processing.

All samples were subjected to SP3 sample preparation[91] on an Agilent BRAVO liquid handling robot. Ten µg of a 1:1 mixture of hydrophilic and hydrophobic carboxyl-coated paramagnetic beads (SeraMag, #24152105050250 and #44152105050250, GE Healthcare) were added for each µg of protein. Protein binding was induced by the addition of acetonitrile to a final concentration of 50% (v/v). Samples were incubated at room temperature for 10 min. The tubes were placed on a magnetic rack, and beads were allowed to settle for 3 min. The supernatant was discarded, and beads were rinsed three times with 200 µL of 80% ethanol without removing the tubes from the rack. Beads were resuspended in digestion buffer containing 50 mM triethylammonium bicarbonate and both Trypsin (Serva) and Lys-C (Wako) in a 1:50 enzyme-to-protein ratio. Protein digestion was performed for 14 h at 37 °C in a PCR cycler. Afterward, the supernatant was recovered and dried down in a vacuum concentrator.

## Proteomics using Mass spectrometry

All samples obtained during the late exponential and transition phase were analyzed on an Orbitrap Exploris (Thermo Scientific) equipped with a FAIMS Pro device and coupled to a 3000 RSLC nano UPLC (Thermo Scientific). Samples were loaded on a PepMap trap cartridge (300 µm i.d. × 5 mm, C18, Thermo) with 2% acetonitrile and 0.1% TFA at a flow rate of 20 µl/min. Peptides were separated over a 50 cm analytical column (Picofrit, 360 µm O.D., 75 µm I.D., 10 µm tip opening, non-coated, New Objective) that was packed in-house with Poroshell 120 EC-C18, 2.7 µm (Agilent). Solvent A consists of 0.1% formic acid in water. Elution was carried out at a constant flow rate of 300 nl/min using a 60-min method: 8–30% solvent B (0.1% formic acid in 80% acetonitrile) within 23 min, 30–45% solvent B within 3 min, 45–98% buffer B within 0.5 min, followed by column washing and equilibration. Data acquisition on the Orbitrap Exploris was carried out using a data-dependent acquisition (DDA) method in positive ion mode. MS survey scans were acquired from 375–1500 m/z in profile mode at a resolution of 60,000. AGC target was set to 200% at a maximum injection time of 25 ms. Peptides with charge states 2–6 were isolated within a window of 1.2 m/z and subjected to HCD fragmentation at a normalized collision energy of 27%. The MS2 AGC target was set to 200%, allowing a maximum injection time of 22 ms. Product ions were detected in the Orbitrap at a resolution of 15,000. Precursors were dynamically excluded for 20 s. The cycle time was set to 1 s for each of the two FAIMS voltages of −45 and −65 V.

LC-MS/MS analysis of the heat shock samples was performed on an Ultimate 3000 UPLC coupled to a Lumos hybrid mass spectrometer (Thermo Scientific). Peptides were loaded onto a PepMap trap cartridge (300 µm i.d. × 5 mm, C18, Thermo) with 2% acetonitrile and 0.1% TFA at a 20 µl/min flow rate. Peptides were separated over a 50 cm

analytical column (Picofrit, 360 µm O.D., 75 µm I.D., 10 µm tip opening, non-coated, New Objective) that was packed in-house with Poroshell 120 EC-C18, 2.7 µm (Agilent). Solvent A consists of 0.1% formic acid in water. Elution was carried out at a constant flow rate of 250 nl/min using a 60-min method: 8–30% solvent B (0.1% formic acid in 80% acetonitrile) within 23 min, 30–45% solvent B within 3 min, 45–98% buffer B within 0.5 min, followed by column washing and equilibration.

The mass spectrometer was operated in data-independent acquisition (DIA) and positive ionization mode. MS1 full scans (375–1500 m/z) were acquired with a resolution of 60,000, a normalized automatic gain control target value of 120%, and a maximum injection time set to 55 ms. Peptide fragmentation was performed using higher energy collision-induced dissociation and a normalized collision energy of 30%. MS2 spectra were acquired with a resolution of 30,000 (first mass 100 m/z) with 8 m/z DIA windows between 400 and 1200 m/z. A normalized automatic gain control target value of 100%, and a maximum injection time of 54 ms.

For pulldown samples, proteins were processed following the SP3 protocol as described above. Label-free DIA analyses of peptides were acquired over 120 min by an Orbitrap Exploris 480 (Thermo Scientific) coupled to a 3000 RSLC nano UPLC (Thermo Scientific). Samples were loaded on a pepmap trap cartridge (300 µm i.d. × 5 mm, C18, Thermo) with 2% acetonitrile, 0.1% TFA at a flow rate of 20 µL/min. Peptides were separated over a 25 cm analytical column (PepSep C18, 75 µm I.D., 1.5 µm). Solvent A consists of 0.1% formic acid in water. Elution was carried out at a constant flow rate of 250 nL/min within 120 min. Initially, a two-step linear gradient was applied: 3–30% solvent B (0.1% formic acid in 80% acetonitrile) within 70.5 min, 30–45% solvent B within 13 min, followed by column washing and equilibration. The column was kept at a constant temperature of 50 °C.

The MS was operated in DIA mode for single-injection quantitative measurements of individual samples with the following settings: 60k MS1 resolution, MS1 scan range 375–1600 m/z, 15k MS2 resolution, MS2 scan range 120–1600 m/z, Normalized AGC target of 1000%, maximum injection time 54 ms, and fixed normalized collision energy of 30. 12 m/z precursor isolation windows with optimized window placements from 400.4319 to 1210.8024 m/z.

## Proteomics data analysis

For samples at the late exponential and transition phase, Raw files were processed with MSFragger v3.5 using the default settings. Briefly, peak lists were extracted from raw files and searched using a Uniprot *Bacillus subtilis* database (v220530, taxonomy ID 224308) and a database containing sequences of common contaminants and decoys. Trypsin/P was set as enzyme specificity, allowing a maximum of two missed cleavages. Peptide length was set from 7 to 50. Precursor and Fragment mass tolerances were set to 10 and 20 ppm, respectively. Peptide matches were filtered to 1% FDR using PeptideProphet and Philosopher (v4.4.0).

For samples during heat shock, DIA-NN (DIA-NN v1.8.1) was employed for DIA data analysis. The main DIA-NN report, "report.txt", was used for all calculations. Fasta file (Bacillus_subtilis_Strain168_UP000001570.fasta) was digested with a maximum of one missed cleavage. Peptide length was restricted from 7 to 30 peptides, and the precursor m/z range was set from 300 to 1800. N-terminal methionine excision was enabled. Cysteine carbamidomethylation was selected as a fixed modification, and methionine oxidation and N-terminal acetylation as variable modifications. The maximum number of variable modifications was set to two. All other parameters were at default settings, including the 1% precursor FDR and enabled match between runs (MBR). The main DIA-NN report, "report.pg_matrix.tsv," was used for all calculations.

For pulldown samples, raw data analysis was performed using Spectronaut® (Biognosys AG, Zurich, Switzerland) version 19.7.250203.62635 in directDIA+ deep mode using a UniProt *B. subtilis*

database (v220530, taxonomy ID 224308). Enzyme specificity was set to "Trypsin/P". Up to two missed trypsin cleavages were allowed. Peptide length was set from 7 to 52. Methionine oxidation and Acetyl (Protein N-term) were set as a variable, and carbamidomethylation on cysteine residues was used as a static modification. Maximum number of variable modifications per peptide was set to five. The FDR for PSM-, peptide-, and protein-level was set to 0.01. All tolerances were set to dynamic for pulsar searches.

The R statistical computing language was used for data processing (v4.4.1). The main comparison of WT, $\Delta ymcA$, (p)ppGpp$^0$, and (p)ppGpp$^0$ $\Delta ymcA$ mutants was based on samples from data-dependent acquisition (DDA) experiments. Raw peptide intensity values from label-free quantitation were imported as tabular output from MSFragger. The MSstats package (v4.12.1)[92] was used to convert raw intensities to an MSstats experiment. In this process, peptide quantification data was log$_2$-transformed, median-normalized, and summarized to protein abundance. All features mapping to a protein were used for quantification, Tukey's median polish was used as a summarization strategy, and missing values were imputed using MSstats' default strategy. Feature abundance and inferred protein quantity were visually inspected using profile plots. Experimental groups were compared using MSstats' *groupComparison* function, and the similarity of conditions and replicates was inspected using principal component analysis (PCA, function *prcomp*). For annotation, protein-related information was fetched from Uniprot using the Uniprot REST API (14 June 2024). In addition to the DDA experiments, a time series of WT and $\Delta ymcA$ mutants was obtained using data-independent acquisition (DIA) proteomics. To quantify relative changes, the log$_2$ fold change of each time point was calculated using the zero time point as a reference. To quantify absolute protein abundance for the ribosomal proteome sector, the coarse-graining strategy was used as in Refs. 93, 94. Briefly, two protein sectors (groups) were created containing all annotated proteins of ribosomal subunits. Raw MS1 intensities for all proteins of the respective group and time points were summed up and divided by the total sum of protein intensities, yielding the protein mass fraction per sector.

### Statistics and reproducibility

Statistical analyses, including Student's t-test, Wilcoxon, and multiple testing adjustments, and most visualizations were performed using R (v 4.2.2) using the packages *ggpubr* (v 0.6.1) and *tidyverse* (v 2.0). Omics data were analyzed using the packages described above. Enrichment analyses were performed using GSEA (v 4.1)[95]. For Supplementary Fig. 6, the ribosome structure was retrieved from PDB (ID: 3J9W)[96,97] and visualized and annotated using ChimeraX[98].

No statistical method was used to predetermine the sample size. Unless stated otherwise, experiments were performed in three biological replicates. This information can be found in the respective figure legend. In the case of the Microscopy-based half-life determinations, the experiment was conducted two times independently. Approximately 100 cells were quantified for each replicate, time point, and strain. The single-cell mean fluorescence intensity values and raw microscopy images are available in the source data. The experiments were not randomized. The Investigators were not blinded to allocation during experiments and outcome assessment, as we performed computational analyses that were insensitive to human bias.

Data exclusions: In Fig. 4C, alternative stress-induced r-proteins were removed for visualization reasons. The changes in their expression are visualized in Figs. 3C and 4B otherwise.

### Reporting summary

Further information on research design is available in the Nature Portfolio Reporting Summary linked to this article.

## Data availability

The Amplicon-seq and RNA-seq raw reads generated in this study have been deposited at the European Nucleotide Archive (ENA) under accession number PRJEB90661. Raw proteomic data have been deposited at the PRIDE database under the accession number PXD065465 for DDA proteomics of late exponential and transition phase samples. The DIA proteomics raw data from the heat shock experiment have been deposited at the PRIDE database under the accession number PXD065462. The YaaT-pulldown proteomic data are accessible using the accession number PXD074523 in PRIDE. The Source data for each figure are provided with this paper. It can also be found in EDMOND [https://doi.org/10.17617/3.E9LOGJ], along with raw microscopy images, processed sequencing data, and the code for Fig. 2D, Supplementary Fig. 2B, and Supplementary Fig. 3. Source data are provided with this paper.

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

## Acknowledgements

We thank Gert Bange (Philipps-Universität Marburg), Mihaela Pruteanu, and Rainer Nikolay (Max Planck Unit for the Science of Pathogens) for critically reading the manuscript and providing valuable feedback.

## Author contributions

K.T., K.D., and F.A.C.: Conceptualization. K.D., F.A.C., K.T., R.A.-B., T.F.W., and K.A.: Methodology. F.A.C., K.D., K.S., V.S., K.H., and F.K.: Investigation. K.D., F.A.C., R.A.-B., M.J., K.A., S.R., and K.T.: Formal analysis. F.A.C., K.D., R.A.-B., M.J.: Visualization. F.A.C., K.D., and K.T.: Writing – Original Draft. K.T. and E.C.: Funding acquisition. All authors: Writing – Review & Editing.

## Funding

This work was funded by the Deutsche Forschungsgemeinschaft (Tu106/8, SPP1879 to K.T. & Leibniz Prize to E.C.) and the Max Planck Society (to E.C. and K.T.). Open Access funding enabled and organized by Projekt DEAL.

## Competing interests

The authors declare no competing interests.
