## [Transparent Peer Review file · Nature Communications]

Stress-induced ribosome degradation in *Bacillus subtilis* is mediated by the RNase Y-specificity complex

Corresponding Author: Professor Kürsad Turgay

Version 0:

Reviewer comments:

Reviewer #1

(Remarks to the Author)

This study reports on the role of the RNase Y-specificity complex (YmcA/YaaT/Ylbf, RicATF) in stress-induced ribosome degradation. The phenotypic analysis of mutants in this complex is interesting and will serve as an excellent starting point for investigating its global physiological role. In particular, the detailed investigation of the changes in transcript cleavage and the RNA-seq data is well performed and analyzed and these results will be significant for the field. However, I do have some points that the authors should address in a revision.

1. The title refers to "Stress-induced" but really the only stress studied is heat. Transition phase is not traditionally thought of as a stress response (as to be contrasted with nutrient starvation).
2. Transition phase, as I understand it, is defined as the often extended period of slower growth between exponential phase (fastest growth) and stationary phase (no growth). In Fig. 2A, both arrows point to what appears to be a similar slope, at least in the in wt growth curve. Please clarify the difference. Also Fig. 4A, it looks like lag phase not exponential is labeled as such.
3. In Fig. 3F, why there is so little 100S in the WT at the "transition phase" point? It should be more specifically described what point on the growth curve the samples are taken. The difference between the WT and delta ymcA could simply be due to differences in the growth curves of the strains as in Fig. 2A.
4. The known dependence of HPF synthesis on (p)ppGpp, complicates interpretation of the effects of (p)ppGpp on ribosome abundance (e.g., Fig. 4). That is, the absence of HPF in a (p)ppGpp null will result in decreased ribosome abundance but a (p)ppGpp null also has increased global protein synthesis, presumably including r proteins. Thus, it is not straightforward to claim as they do that "These experiments suggest that the Y-complex and (p)ppGpp control ribosome levels and translation independently during post-exponential growth." (line 250).
5. In addition, there is evidence concerning the interaction of RNases with (p)ppGpp, (e.g., *E. coli* RppH, PMID: 31960065), which should be mentioned in the Discussion and may be relevant to understanding the relationship between these two systems (see #4, above).
6. Fig. 6 indicates that (p)ppGpp inhibits transcription by reducing GTP levels but this is inconsistent with studies that do not observe a decrease in GTP during transition phase, even as (p)ppGpp levels increase (PMID: 28887466, 40845056).

Reviewer #2

(Remarks to the Author)

This manuscript describes a previously uncharacterized mechanism for translational control and resource allocation in *Bacillus subtilis* during stress response. Through a genetic screen designed to identify factors involved in the heat shock response, the authors identified the Y-complex (RicAFT complex), which confers specificity to the endonuclease RNase Y, as an important contributor under stress conditions such as heat exposure or transition to stationary phase.

The study presents evidence that the Y-complex functions as a regulatory hub influencing gene expression, protein synthesis, and resource allocation. The results indicate that the Y-complex and RNase Y initiate degradation of rRNAs from mature ribosomes (23S and 16S rRNA), leading to a reduction in ribosome abundance during nutrient limitation or heat shock. This degradation involves cleavages at defined regions within rRNA secondary structures.

The physiological relevance of this mechanism is supported by observations that mutants lacking a Y-complex component exhibit elevated ribosome levels and increased accumulation of aggregated proteins following heat shock, thereby imposing

additional demands on the protein quality control (PQC) system. The data further suggest that the Y-complex system acts independently of, yet in coordination with, the alarmone (p)ppGpp to regulate ribosome abundance and translation. Overall, the manuscript provides insight into how *B. subtilis* coordinates ribosome biogenesis (via (p)ppGpp) and active ribosome turnover (via the Y-complex/RNase Y system) to maintain cellular homeostasis during environmental stress. The approach used in this study is valid. The qualities of data are good. There is an appropriate use of statistics. References are appropriate to credit previous works. Abstract and introduction are clear. The manuscript is well presented.

Reviewer #3

(Remarks to the Author)

The manuscript of Cornejo et al employs a genetic screen to search for novel factors involved in heat shock in *Bacillus subtilis*, leading to the identification of the Y-complexes, as an important player. The manuscript provides a comprehensive investigation into the role of the Y-complex—comprising YmcA, YlbF, and YaaT—in regulating ribosome levels and processing RNA during stress responses in *B. subtilis*. Using a combination of genetic screenings, RNA-seq, northern blot analyses, mass spectrometry, and structural mapping, the authors demonstrate that the Y-complex interacts with RNase Y to mediate the degradation of rRNA, particularly in response to heat shock and nutrient depletion. They further elucidate the relationship between the Y-complex and the global stress alarmone (p)ppGpp, highlighting their independent yet converging roles in controlling ribosome abundance and cellular proteostasis. The work advances our understanding of post-transcriptional regulation during bacterial stress, particularly in the context of ribosome turnover, and suggests that the Y-complex is a key regulatory hub modulating gene expression and resource allocation in *B. subtilis*.

Overall, this manuscript presents significant advances in understanding bacterial ribosome regulation under stress. It convincingly establishes the Y-complex as a central player in RNA processing and ribosome degradation, with broad implications for bacterial adaptation and survival. The work adds valuable mechanistic insights into post-transcriptional regulation of ribosomes, with potential relevance across bacterial species. In principle, the manuscript is already in an acceptable form, however, I think it would be improved by addressing some of the points below:

1. The study links the Y-complex to heat-shock survival using two orthogonal assays: a pooled barcoded fitness screen at 50°C (workflow and outcome shown in Figure 1A–B) and independent time-kill curves at 54°C demonstrating reduced viability for *ymcA*, *ylbF*, and *yaaT* deletions (Figure 1C). These data strongly support the association between Y-complex loss and heat sensitivity. However, the study does not include genetic complementation demonstrating that reintroduction of *ymcA*, *ylbF*, or *yaaT* restores heat-shock survival to the phenotype observed in Figure 1B–C. This limits causal attribution of the survival defect specifically to Y-complex loss. Demonstrating rescue in the same background would solidify that the viability drop at 54°C (Figure 1C) and fitness loss at 50°C (Figure 1A–B) are directly due to Y-complex disruption.
2. The transcriptome-wide RNA end mapping compares wild type and a *ymcA* deletion at late exponential and transition phases (Figures 2A–2B). The analysis finds many unique and stepped ends with stabilization patterns and localizes these across ORFs and UTRs (Figures 2C–2F; Supplementary Figures 2A and 3A–F), culminating in pathway enrichment that emphasizes metabolism and translation (Figure 2G; Supplementary Figure 4A). These data underpin the claim that the complex processes specific categories of transcripts. The study does not include genetic complementation of the *ymcA* deletion to demonstrate that the observed loss of cleavage sites and pathway enrichment are specifically due to Y-complex absence. Without complementation, off-target or pleiotropic effects of the deletion cannot be excluded, especially given pleiotropic phenotypes reported for Y-complex mutants.
3. The rRNA end-mapping in Figure 3A, together with the structural localization in Supplementary Figures 5A/5B and the 3D mapping in Supplementary Figure 6, is used to argue that the Y-complex generates specific cleavage sites in 23S and 16S rRNA; northern blots in Figure 3B corroborate degradation intermediates. The RNA end-mapping pipeline uses random assignment for multi-mapped reads and lacks replicate-level reproducibility metrics, risking false-positive rRNA cleavage site calls. Literature confirms that random assignment of multi-mapped reads 'can lead to false positives in differential expression analysis if not correctly assigned' [1], with rRNA depletion specifically increasing multi-mapping rates that affect quantification accuracy [2]. rRNA contains repetitive and highly structured regions where reads frequently multi-map; using STAR in random-best mode without EM-weighting or unique-only constraints can create artificial end clusters. Without an IDR-style reproducibility assessment across replicates [3] and an end-focused normalization that accounts for rRNA depletion efficiency, the peaks in Figure 3A and their proximity to functional regions in Supplementary Figure 6 may reflect mapping/preprocessing artifacts rather than true cleavage.
4. While the RNA processing data suggest the Y-complex guides RNase Y to specific cleavage sites, the precise molecular mechanism of substrate recognition remains somewhat speculative. The evidence for initiation of rRNA decay includes Y-complex-dependent cleavage sites within 23S and 16S rRNA (Figure 3A; Supplementary Figures 5A–B and 6), reduced 23S/16S degradation intermediates in the *ymcA* deletion (Figure 3B; Supplementary Figure 7A), and recovery of RNase Y together with ribosomal proteins in a YaaT-FLAG pulldown (Supplementary Figure 7B). These observations coherently support rRNA decay guided by the Y-complex and are persuasive. However, the catalytic role of RNase Y in executing these rRNA cleavages is inferred rather than directly demonstrated. The study does not directly test RNase Y catalytic dependency for the observed rRNA cleavages and ribosome decay *in vivo*. Attribution of rRNA cleavage to RNase Y rests on the loss of Y-complex-dependent ends and associations in Supplementary Figure 7B, but no RNase Y catalytic mutant, conditional depletion, or acute inhibition is used to show loss of the cleavage bands seen in Figure 3B and Supplementary Figure 7A. Without a direct RNase Y dependency test, the cleavages mapped in Figure 3A and the pulldown associations could be consistent with other endonucleases initiating decay, leaving the initiating nuclease unresolved.
5. Ribosome abundance changes are inferred from proteomics showing reduced ribosomal protein levels in wild type but impaired reduction in the *ymcA* deletion during transition (Figure 3C), unchanged rRNA and r-protein promoter activities in both strains (Figure 3D–E; Supplementary Figures 7C–E), and sedimentation profiles indicating 100S accumulation in the

ymcA deletion (Figure 3F). These data strongly suggest altered ribosome homeostasis linked to Y-complex function. However, they do not explicitly separate decay of existing ribosomes from changes in biogenesis and assembly dynamics. The experiments do not directly quantify ribosome turnover rates to distinguish active decay of mature ribosomes from altered biogenesis or assembly states. Promoter assays in Figure 3D-E and Supplementary Figures 7C-E address transcriptional control but not rRNA synthesis rates, assembly efficiency, or ribosome half-lives. Proteomic mass fractions in Figure 3C reflect net abundance but cannot resolve whether changes arise from degradation versus assembly alterations; sedimentation (Figure 3F) indicates sequestration into 100S but does not quantify 70S turnover. Without pulse-chase labeling of rRNA or ribosome-specific turnover assessments, attributing abundance changes primarily to Y-complex-initiated decay remains an inference.

6. Support for functional specificity rests on the overrepresentation analysis showing enrichment of Y-complex-dependent cleavages in metabolism and information-processing categories (Figure 2G; Supplementary Figure 4A). In the same dataset, there is a clear preference for cleavages in polycistronic transcripts and internal UTRs (Figure 2E-F; Supplementary Figure 2C). The authors also show that transcript length and abundance alone do not predict cleavage (Supplementary Figure 2D-E), which is reassuring. The enrichment analysis does not control for operon architecture and other structural covariates that could confound the apparent functional specificity. Because the data show a pronounced bias toward polycistronic mRNAs (Figure 2E) and multiple cleavages per polycistronic transcript (Supplementary Figure 2C), functional categories that are predominantly operonic (such as metabolic and translation operons) are more likely to appear enriched in Figure 2G. Although length and abundance were examined (Supplementary Figure 2D-E), the analysis does not stratify by polycistronic versus monocistronic organization, UTR frequency, motif density, or local folding energy, leaving open the possibility that the observed enrichment reflects architecture rather than function. Without a multivariate or stratified enrichment framework controlling for these structural covariates, the conclusion of functional specificity remains partially confounded by transcript architecture.

Version 1:

Reviewer comments:

Reviewer #1

(Remarks to the Author)

The authors have answered my queries to my satisfaction and are to be commended for their serious attention to them.

Reviewer #3

(Remarks to the Author)

The authors have very thoroughly addressed all my concerns and I congratulate them on a great piece of work.

REVIEWER COMMENTS

Please use another color for writing the answers

We thank the reviewers for their valuable comments. We have implemented the requested changes, which have strengthened the manuscript. In this response to the reviewers, we have included figures to support our points; however, not all of them have been included in the manuscript. Here, we summarize the figures that have been added in the new version:

- 100S formation even in the absence of (p)ppGpp during the transition phase (**Fig. R1.2**) as new **Supplementary Fig. 8**
- Complementation of $\Delta ymcA$ in heat shock sensitivity and growth curves (**Fig. R3.1**) as panels **C** and **D** in **Supplementary Fig. 1**
- Growth defects of an RNase Y conditional depletion mutant and reduced rRNA degradation in this mutant (**Fig. R3.3**) as panels **C** and **D** in **Supplementary Fig. 7**
- Measurements of ribosome decay using microscopy-based pulse-chase assay (**Fig. R3.4**) as panels **F** and **G** in **Fig. 3**

Other changes to the figures include updating **Supplementary Figures 5** and **6** to address mapping of RNA ends in repetitive and conserved sequences. We have also updated the model in **Figure 6** to address inconsistencies in GTP depletion. Finally, we have included independent replicates in our YaaT-pulldown analysis in **Supplementary Fig. 7B**. We have made minor edits to the text to improve readability, which are also highlighted in the revised version.

Reviewer #1 (Remarks to the Author):

This study reports on the role of the RNase Y-specificity complex (YmcA/YaaT/YibF, RicATF) in stress-induced ribosome degradation. The phenotypic analysis of mutants in this complex is interesting and will serve as an excellent starting point for investigating its global physiological role. In particular, the detailed investigation of the changes in transcript cleavage and the RNA-seq data is well performed and analyzed and these results will be significant for the field.

Thank you for the positive comments.

However, I do have some points that the authors should address in a revision.

1. The title refers to “Stress-induced” but really the only stress studied is heat. Transition phase is not traditionally thought of as a stress response (as to be contrasted with nutrient starvation).

We thank the reviewer for the opportunity to clarify our terminology.

Heat stress, like oxidative or salt stress, is a protein-unfolding stress that disrupts protein homeostasis. Bacterial cells respond to stress by inducing chaperones and downregulating translation to reduce the load on the protein quality control system. As shown in our previous studies, the stringent and heat shock responses share stress-response mechanisms, specifically the downregulation of translation (Schäfer *et al.*, 2019, 2020). Even developmental processes such as competence development, which begin after entry into the stationary phase, utilize (p)ppGpp-mediated downregulation of translation (Hahn *et al.* 2015).

Similarly, nutrient starvation, which occurs during the transition to the stationary phase, induces stress-response mechanisms that significantly overlap with protein-unfolding stress responses,

including downregulation of translation. In this context, we consider (p)ppGpp to be a globally relevant secondary messenger signaling utilized across several stress conditions.

Our data show that Y-complex – RNase Y mediated ribosome degradation was not only occurring during proteotoxic stress but also during nutritional stress (**Fig. 3-5**), which is important for limiting growth and protein synthesis.

We reason that the transition to the stationary phase represents a mild physiological stress in which limiting nutrients are slowly depleted. This triggers several well-known stress response pathways, including the stringent response, the CodY-derepression (Hecker and Völker, 1990; Geiger and Wolz, 2014), and the general stress response (Völker et al., 1995). These pathways form an intricate interconnected regulatory network that modulates, interferes, and interacts with more global processes such as transcription, RNA stability, translation, and protein activity and stability.

Stress can therefore be understood in a broader term, describing situations that perturb the steady-state growth of bacteria, prompting the activation of several homeostatic mechanisms (stress responses) to control these global and specific mechanisms (Martin, 2014).

To reflect our rationale, we have added the following sentence to lines L103-106: “Such a post-exponential growth change in the transition to the stationary phase can be considered a physiological stress where nutrients are depleted slowly from the media. Thereby, activating the general stress response and stringent response, including the CodY regulon²⁹⁻³¹.”

References:

- Geiger, T., & Wolz, C. (2014). Intersection of the stringent response and the CodY regulon in low GC Gram-positive bacteria. *International Journal of Medical Microbiology*, 304(2), 150-155.
- Hahn, J., Tanner, A. W., Carabetta, V. J., Cristea, I. M., & Dubnau, D. (2015). ComGA-RelA interaction and persistence in the *Bacillus subtilis* K-state. *Molecular Microbiology*, 97(3), 454-471.
- Hecker, M., & Völker, U. (1990). General stress proteins in *Bacillus subtilis*. *FEMS Microbiology Letters*, 74(2-3), 197-213.
- Martin, C. (2014). What is stress? *Current Biology*, 24(10), R403-R405
- Schäfer, H., Beckert, B., Frese, C. K., Steinchen, W., Nuss, A. M., Beckstette, M., ... & Turgay, K. (2020). The alarmones (p)ppGpp are part of the heat shock response of *Bacillus subtilis*. *PLoS genetics*, 16(3), e1008275.
- Schäfer, H., Heinz, A., Sudzinová, P., Voß, M., Hantke, I., Krásný, L., & Turgay, K. (2019). Spx, the central regulator of the heat and oxidative stress response in *B. subtilis*, can repress transcription of translation-related genes. *Molecular Microbiology*, 111(2), 514-533
- Völker, U., Völker, A., Maul, B., Hecker, M., Dufour, A., & Haldenwang, W. G. (1995). Separate mechanisms activate sigma B of *Bacillus subtilis* in response to environmental and metabolic stresses. *Journal of bacteriology*, 177(13), 3771-3780

2. Transition phase, as I understand it, is defined as the often extended period of slower growth between exponential phase (fastest growth) and stationary phase (no growth). In Fig. 2A, both arrows point to what appears to be a similar slope, at least in the in wt growth curve. Please clarify the difference. Also Fig, 4A, it looks like lag phase not exponential is labeled as such.

We sincerely thank you for raising this important point. We agree that the growth phases can be difficult to distinguish in the original figures.

The growth curves in Fig. 2A and 4A are plotted on a linear scale to better visualize the growth defect of the mutants during the transition phase. On the linear scale, the exponential growth appears compressed, including points that appear to be from the lag phase. To clarify this point, we provide here a side-by-side comparison of the data from Fig. 2A on both linear and log scales (**Figure R1.1**). When viewed on a semi-log plot, the exponential phase for the WT and mutants ends around OD 1 (approximately 180 min after inoculation). The transition phase we describe starts after this point, as the growth slows down before the culture reaches the true stationary phase.

We thank the reviewer for pointing out that we have mistakenly labelled part of the lag phase as the exponential phase. We have corrected this in a revised version of **Figures 2A** and **4A**. We have also added to the Figure legends that the growth phases were classified based on semi-log plots (L861 and L907).

Figure R1.1 Growth curve of WT, *ymcA*, *yIbF*, and *yaaT* deletion mutants in LB at 37°C plotted in linear (*left*) or log₂ (*right*) scale. The blue bar marks the exponential phase. The collection points for late exponential and transition phase are shown with a pink arrow. The mean and SD of three biological replicates are displayed.

3. In Fig. 3F, why there is so little 100S in the WT at the “transition phase” point? It should be more specifically described what point on the growth curve the samples are taken. The difference between the WT and delta *ymcA* could simply be due to differences in the growth curves of the strains as in Fig. 2A.

Regarding the sampling time point, we hope that our response to comment 2 clarifies these concerns. In the log₂-scaled growth curve, it is noteworthy that although the Y-complex mutants show a growth defect at the transition phase, they are in the same growth phase as the WT.

Regarding the low or absent presence of 100S in the WT, very few studies focus on the “transition phase”; therefore, there is little evidence for the amount of 100S hibernating ribosomes in *B. subtilis* WT at this growth phase. We acknowledge that transcript levels of *hpf*

increase in a (p)ppGpp-dependent manner during the transition phase (Nagarajan et al., 2025) (**more discussion on this point in the following response**). Given that Hpf can also be detected in 70S by Western blotting (Feaga et al., 2020), it is possible that our time point in the transition phase is too early to detect hibernating ribosomes in the WT. Together with the higher levels of ribosomes, we believe that the increase in the number of hibernating ribosomes is a consequence of excess ribosomes rather than differences in growth phase.

We have labeled the corresponding time points in the growth curve in **Figure 2A** in more detail and added this information to the legend of **Figure 3H**, L899: “Ribosome sedimentation profile in 10-50% sucrose gradients of WT and $\Delta ymcA$ in exponential (OD 0.3), late exponential, and transition phases (as shown in Fig. 2A).”

4. The known dependence of HPF synthesis on (p)ppGpp, complicates interpretation of the effects of (p)ppGpp on ribosome abundance (e.g., Fig. 4). That is, the absence of HPF in a (p)ppGpp null will result in decreased ribosome abundance but a (p)ppGpp null also has increased global protein synthesis, presumably including r proteins. Thus, it is not straightforward to claim as they do that “These experiments suggest that the Y-complex and (p)ppGpp control ribosome levels and translation independently during post-exponential growth.” (line 250).

Thank you for highlighting this very interesting point. The transcription of the gene *hpf* is also regulated by SigB, SigH, and CodY (Belitsky & Sonenshein, 2013; Drzewiecki et al., 1998). Previous studies have shown that the accumulation of (p)ppGpp leads to the formation of 100S ribosomes (Tagami et al., 2012) and that the transcription of *hpf* is (p)ppGpp-dependent via SigH during the transition phase (Nagarajan et al., 2025). However, given that the general stress response (SigB) is also activated during the transition phase (response to point 1), we evaluated the Hpf protein levels using our proteomic data from the transition phase. Hpf is highly upregulated when cells shift into the transition phase (**Figure R1.2A**). This upregulation also occurs in the (p)ppGpp⁰, $\Delta ymcA$, and (p)ppGpp⁰ $\Delta ymcA$, suggesting an (p)ppGpp-independent control of Hpf levels. Furthermore, Hpf levels are comparable between WT, $\Delta ymcA$, (p)ppGpp⁰, and the double mutant at the transition phase, with slightly higher values in the (p)ppGpp⁰ strain (**Figure R1.2B**).

We wondered whether these Hpf levels during the transition phase were sufficient to allow 100S formation in each strain, given that the (p)ppGpp⁰, $\Delta ymcA$, and double mutant strains accumulate ribosomes at this time point. To this end, we calculated the difference in Hpf levels relative to the average r-protein level for each strain and condition (**Figure R1.2C**). At the late exponential phase, Hpf levels are lower than r-protein levels, but at the transition phase, Hpf levels are comparable to r-protein levels, suggesting that there is enough Hpf to form 100S for each ribosome. Next, we tested 100S formation in the (p)ppGpp⁰ background (**Figure R1.2D**). At the transition phase, (p)ppGpp⁰ strain shows a small accumulation of 100S, which is stronger in the (p)ppGpp⁰ $\Delta ymcA$ strain. This suggests a (p)ppGpp-independent 100S formation pathway that may be triggered by other transcription factors that regulate *hpf*, most probably SigB. Interestingly, Hpf levels in the transition phase are enough for 100S formation on every ribosome. We believe that another trigger or licensing factor may be necessary to fully induce 100S formation; however, this is beyond the scope of our current manuscript.

We agree with the reviewer’s point that a (p)ppGpp⁰ strain is more translationally active than the WT at the transition phase, and that this might also involve increased synthesis of r-proteins, and thus, ribosomes. To address the reviewer’s point, we have changed our statement that both

systems [(p)ppGpp and the Y-complex] function independently to control ribosome levels to indicate that both systems are being critical for this process:

We have included these results (new **Supplementary Fig. 8**) and modified the statements that the Y-complex and (p)ppGpp independently control ribosome levels and translation in L281-288:

“Given the increased accumulation of ribosomes in the (p)ppGpp⁰ *ΔymcA* strain, we evaluated whether 100S ribosomes could also form in this strain during the transition phase. We observed a strong increase in Hpf levels during the transition phase, including in the (p)ppGpp⁰ and (p)ppGpp⁰ *ΔymcA* strains (**Supplementary Fig. 8A-B**). These elevated Hpf levels are similar in quantity to those of r-proteins, suggesting that they could sustain 100S ribosome formation (**Supplementary Fig. 8C**). Indeed, we observed accumulation of 100S ribosomes in the (p)ppGpp⁰ and (p)ppGpp⁰ *ΔymcA* strains (**Supplementary Fig. 8D**), suggesting the presence of a (p)ppGpp-independent pathway of 100S formation.”

The more severe growth defect caused by the absence of (p)ppGpp and the Y-complex, combined with the reduced downregulation of ribosome levels in the (p)ppGpp⁰ *ΔymcA* strain, suggests that both systems are critical for regulating ribosome levels during post-exponential growth.”

Figure R1.2 100S ribosome formation in (p)ppGpp⁰ strains suggests an additional alarmone-independent mechanism of ribosome hibernation. **A)** Volcano plot of proteomic changes of WT, $\Delta ymcA$, (p)ppGpp⁰, and (p)ppGpp⁰ $\Delta ymcA$ cultures comparing transition to exponential phase. The Hpf protein is highlighted in magenta. The data shows the average of four biological replicates. **B)** Hpf levels measured by mass spectrometry at the transition phase in the mentioned strains. **C)** Hpf levels compared to r-proteins. The Z-score of r-proteins was calculated for each strain and time point independently. Hpf levels were Z-scored using the average and SD of the r-proteins. The boxplot represents the interquartile range (IQR) and the median in the center. Whiskers show the variability outside quartile 1 (Q1) and Q3 and were calculated as $Q1-1.5 \cdot IQR$ and $Q5+1.5 \cdot IQR$, respectively. **D)** Ribosome sedimentation profile in 10-50% sucrose gradients of (p)ppGpp⁰ and (p)ppGpp⁰ $\Delta ymcA$ strains in the transition phase. The 70S and 100S ribosomes are highlighted with a blue or red box, respectively.

5. In addition, there is evidence concerning the interaction of RNAses with (p)ppGpp, (e.g., *E. coli* RppH, PMID: 31960065), which should be mentioned in the Discussion and may be relevant to understanding the relationship between these two systems (see #4, above).

We thank the reviewer for this suggestion. We have added it to the discussion in L405-409, as follows:

“Interestingly, (p)ppGpp has been shown to directly modulate the activity of RNases in other species, such as the Nudix hydrolase RppH in *E. coli* and PNPase in *Streptomyces*^{63,64}, linking (p)ppGpp to RNA metabolism. Thereby, the alarmones can be involved in rRNA degradation via the Y-complex and RNase Y, or another RNase.”

6. Fig. 6 indicates that (p)ppGpp inhibits transcription by reducing GTP levels but this is inconsistent with studies that do not observe a decrease in GTP during transition phase, even as (p)ppGpp levels increase (PMID: 28887466, 40845056).

We acknowledge that the published results indicate some inconsistency between the reduction in GTP during the transition phase and that observed under experimentally induced amino acid starvation. This can be rooted in an acute drop in amino acid levels during downshift experiments, compared with the gradual depletion during growth phase transitions.

Regarding the references provided, we note that Varik et al. (2017) observed a drop in GDP levels when ppGpp levels increased during the transition phase in *E. coli* (Fig. 4A of their article). This could be explained by a preferential use of GDP to synthesize ppGpp over GTP for pppGpp. In Hydorn et al. (2025), the authors observed a modest decline in GTP levels (Fig. S3 of that article). However, a time delay between the peak of (p)ppGpp and GTP depletion might also occur, as (p)ppGpp affects GTP biosynthesis at multiple levels (salvage pathways, *de novo* biosynthesis, and inhibition of PurR regulon transcription).

In *B. subtilis*, the regulation of ribosome synthesis differs from the *E. coli* model; here, the concentration of the initiating nucleotide (GTP) was identified as a direct limiting factor for the transcription of rRNA and r-proteins, as demonstrated by Krásný & Gourse (2004) and Krásný et al. (2008). To ensure that our model reflects the possible inconsistencies observed in the drop of GTP levels, we have modified the arrow going from (p)ppGpp to transcription, which now reads: “Inhibition of transcription by reduced GTP levels”. We have also updated the figure legend as follows:

“Figure 6. Control of ribosome levels by (p)ppGpp and the Y-complex. The alarmone (p)ppGpp reduces ribosome biogenesis by indirectly inhibiting the transcription of rRNA and r-protein mRNAs. This indirect inhibition occurs in promoters containing a G as the initiating nucleotide (+1), being then sensitive to changes in GTP concentration in the cell¹⁸. This connection is shown with a dotted line as GTP levels have been shown to drop after (p)ppGpp biosynthesis during amino acid starvation or other starvation-inducing conditions, but recent reports show mild or no decrease in GTP levels at the transition phase⁹¹. (p)ppGpp also inhibits GTPases that are involved in new ribosome assembly. Additionally, (p)ppGpp reduces translation by binding to specific GTPases. The Y-complex can initiate the degradation of ribosomes, which need to be transported to the membrane by unknown mechanisms. Exonucleases and AAA+ protease complexes might assist in ribosome degradation for the complete degradation of rRNA and r-proteins, which could partially replenish the nucleotide and amino acid pools during starvation.”

References

Hydorn, M., Nagarajan, S. N., Fones, E., Harwood, C. S., & Dworkin, J. (2025). Analysis of (p)ppGpp metabolism and signaling using a dynamic luminescent reporter. *PLoS Genetics*, 21(8), e1011691

Krásný, L., & Gourse, R. L. (2004). An alternative strategy for bacterial ribosome synthesis: *Bacillus subtilis* rRNA transcription regulation. *The EMBO journal*, 23(22), 4473.

Krásný, L., Tišerová, H., Jonák, J., Rejman, D., & Šanderová, H. (2008). The identity of the transcription+ 1 position is crucial for changes in gene expression in response to amino acid starvation in *Bacillus subtilis*. *Molecular microbiology*, 69(1), 42-54.

Varik, V., Oliveira, S. R. A., Hauryliuk, V., & Tenson, T. (2017). HPLC-based quantification of bacterial housekeeping nucleotides and alarmone messengers ppGpp and pppGpp. *Scientific reports*, 7(1), 11022.

Reviewer #2 (Remarks to the Author):

*This manuscript describes a previously uncharacterized mechanism for translational control and resource allocation in *Bacillus subtilis* during stress response. Through a genetic screen designed to identify factors involved in the heat shock response, the authors identified the Y-complex (RicAFT complex), which confers specificity to the endonuclease RNase Y, as an important contributor under stress conditions such as heat exposure or transition to stationary phase.*

The study presents evidence that the Y-complex functions as a regulatory hub influencing gene expression, protein synthesis, and resource allocation. The results indicate that the Y-complex and RNase Y initiate degradation of rRNAs from mature ribosomes (23S and 16S rRNA), leading to a reduction in ribosome abundance during nutrient limitation or heat shock. This degradation involves cleavages at defined regions within rRNA secondary structures.

The physiological relevance of this mechanism is supported by observations that mutants lacking a Y-complex component exhibit elevated ribosome levels and increased accumulation of aggregated proteins following heat shock, thereby imposing additional demands on the protein quality control (PQC) system. The data further suggest that the Y-complex system acts independently of, yet in coordination with, the alarmones (p)ppGpp to regulate ribosome abundance and translation.

*Overall, the manuscript provides insight into how *B. subtilis* coordinates ribosome biogenesis (via (p)ppGpp) and active ribosome turnover (via the Y-complex/RNase Y system) to maintain cellular homeostasis during environmental stress.*

The approach used in this study is valid. The qualities of data are good. There is an appropriate use of statistics.

References are appropriate to credit previous works.

Abstract and introduction are clear. The manuscript is well presented.

We thank the reviewer for their positive assessment of our work.

Reviewer #3 (Remarks to the Author):

*The manuscript of Cornejo et al employs a genetic screen to search for novel factors involved in heat shock in *Bacillus subtilis*, leading to the identification of the Y-complexes, as an important player. The manuscript provides a comprehensive investigation into the role of the Y-complex—comprising YmcA, YlbF, and YaaT—in regulating ribosome levels and processing RNA during stress responses in *B. subtilis*. Using a combination of genetic screenings, RNA-seq, northern blot analyses, mass spectrometry, and structural mapping, the authors demonstrate that the Y-complex interacts with RNase Y to mediate the degradation of rRNA, particularly in response to heat shock and nutrient depletion. They further elucidate the relationship between the Y-complex and the global stress alarmone (p)ppGpp, highlighting their independent yet converging roles in controlling ribosome abundance and cellular proteostasis. The work advances our understanding of post-transcriptional regulation during bacterial stress,*

particularly in the context of ribosome turnover, and suggests that the Y-complex is a key regulatory hub modulating gene expression and resource allocation in *B. subtilis*.

Overall, this manuscript presents significant advances in understanding bacterial ribosome regulation under stress. It convincingly establishes the Y-complex as a central player in RNA processing and ribosome degradation, with broad implications for bacterial adaptation and survival. The work adds valuable mechanistic insights into post-transcriptional regulation of ribosomes, with potential relevance across bacterial species.

Thank you for the positive comments.

In principle, the manuscript is already in an acceptable form, however, I think it would be improved by addressing some of the points below:

1. The study links the Y-complex to heat-shock survival using two orthogonal assays: a pooled barcoded fitness screen at 50°C (workflow and outcome shown in Figure 1A–B) and independent time-kill curves at 54°C demonstrating reduced viability for *ymcA*, *ylbF*, and *yaaT* deletions (Figure 1C). These data strongly support the association between Y-complex loss and heat sensitivity. However, the study does not include genetic complementation demonstrating that reintroduction of *ymcA*, *ylbF*, or *yaaT* restores heat-shock survival to the phenotype observed in Figure 1B–C. This limits causal attribution of the survival defect specifically to Y-complex loss. Demonstrating rescue in the same background would solidify that the viability drop at 54°C (Figure 1C) and fitness loss at 50°C (Figure 1A–B) are directly due to Y-complex disruption.

Complementation analyses are very useful for ruling out polar effects caused by gene deletions. Given that *ymcA*, *ylbF*, and *yaaT* are encoded in different operons across the *B. subtilis* chromosome, and all mutants showed a comparable phenotype, we ruled out polar effects as the cause of the observed heat-shock and growth-defect phenotype.

Nevertheless, we agree that a complementation experiment could corroborate our conclusions. To this end, we complemented our delta *ymcA* strain by reintroducing a copy of the *ymcA* gene under IPTG control at the *amyE* locus. The complemented strain recovered the WT growth phenotype and showed similar survival at 54 °C (Fig. R3.1). This was independent of IPTG addition, suggesting that the leakiness from the *hyperspank* promoter was sufficient to restore the phenotype.

Figure R3.1 Complementation of the *ymcA* deletion rescues the growth defect in LB and survival at 54 °C.

We added this data in **Supplementary Fig. 1C-D** and added the following to L93-96: “In addition, the heat shock sensitivity of the $\Delta ymcA$ strain can be rescued by ectopically expressing *ymcA* from the *amyE* locus, even without inducer, due to the known leakage of the *hyperspank* promoter²⁸ (**Supplementary Fig. 1C**).” and L102-103: “This growth phenotype of the $\Delta ymcA$ strain is completely rescued in a *ymcA*-complemented strain (**Supplementary Fig. 1D**)”

2. *The transcriptome-wide RNA end mapping compares wild type and a ymcA deletion at late exponential and transition phases (Figures 2A–2B). The analysis finds many unique and stepped ends with stabilization patterns and localizes these across ORFs and UTRs (Figures 2C–2F; Supplementary Figures 2A and 3A–F), culminating in pathway enrichment that emphasizes metabolism and translation (Figure 2G; Supplementary Figure 4A). These data underpin the claim that the complex processes specific categories of transcripts. The study does not include genetic complementation of the ymcA deletion to demonstrate that the observed loss of cleavage sites and pathway enrichment are specifically due to Y-complex absence. Without complementation, off-target or pleiotropic effects of the deletion cannot be excluded, especially given pleiotropic phenotypes reported for Y-complex mutants.*

We hope that our response to the previous comment addresses the reviewer's concerns regarding off-target or polar effects in the $\Delta ymcA$ strain, as the complemented strain fully restores the WT phenotype (new **Supplementary Figure 1C-D**).

Regarding the mapping of RNA ends, we used the $\Delta ymcA$ mutant as a representative of the Y-complex. Our decision to use only one strain is based on the previous work from DeLoughery *et al.* (2016). In this study, they demonstrate that the loss of Y-complex-dependent cleavage sites is consistent regardless of whether *ymcA*, *ybfF*, or *yaaT* is deleted. This consistency across the three members of the Y-complex provides a strong internal genetic validation, because it is highly unlikely that independent deletions in three distinct genes, encoded in different operons, would yield the same off-targets. Furthermore, our new data, using conditional depletion of RNase Y (response to comment number 4), mirror the $\Delta ymcA$ defects in rRNA processing/degradation, independently confirming our approach to identifying cleavage sites.

Regarding our assertion that the Y-complex processes specific categories of transcripts, we address this point in detail in our response to the reviewer's 6th comment.

3. *The rRNA end-mapping in Figure 3A, together with the structural localization in Supplementary Figures 5A/5B and the 3D mapping in Supplementary Figure 6, is used to argue that the Y-complex generates specific cleavage sites in 23S and 16S rRNA; northern blots in Figure 3B corroborate degradation intermediates. The RNA end-mapping pipeline uses random assignment for multi-mapped reads and lacks replicate-level reproducibility metrics, risking false-positive rRNA cleavage site calls. Literature confirms that random assignment of multi-mapped reads 'can lead to false positives in differential expression analysis if not correctly assigned' [1], with rRNA depletion specifically increasing multi-mapping rates that affect quantification accuracy [2]. rRNA contains repetitive and highly structured regions where reads frequently multi-map; using STAR in random-best mode without EM-weighting or unique-only constraints can create artificial end clusters. Without an IDR-style reproducibility assessment across replicates [3] and an end-focused normalization that accounts for rRNA depletion efficiency, the peaks in Figure 3A and their proximity to functional regions in Supplementary Figure 6 may reflect mapping/preprocessing artifacts rather than true cleavage.*

We confirm that our method for identifying differential RNA ends relies on replicate data. In our case, we used three biological replicates of the WT and $\Delta ymcA$. Rather than using an IDR-style reproducibility assessment, which was originally developed and tested to estimate the reliability of peak signals in ChIP-seq, we used *edgeR*. This tool integrates replicate information into its statistical model by calculating dispersion estimates to account for replicate variance prior to hypothesis testing. By using this approach, we ensure that our detected RNA ends are only identified when they show a significant change between replicates, with an absolute log2 fold change ≥ 1 and an FDR < 0.05 .

To make our analysis more stringent, after the *edgeR* analysis, we performed additional filtering based on the cleavage ratio (proportion of ends) and the ratio between WT and $\Delta ymcA$ cleavage ratios, and we applied an additional cutoff based on the distribution of cleavage ratios between WT and $\Delta ymcA$.

Regarding multi-mapping, our RNA end-mapping pipeline uses the STAR aligner with the ‘random best assignment’ option, which assigns reads randomly to one of the highest-scoring mapping locations. This should reduce or prevent the artificial accumulation of “clusters” of reads on a single copy of the rRNA, contrary to what the reviewer suggests.

Nevertheless, we further tested whether randomly distributed multi-mapping reads introduce a bias in repetitive genomic regions, such as 23S rRNA regions. First, we tested whether it was possible to use uniquely mapped reads to the *B. subtilis* genome while still identifying cleavage sites in 23S rRNA. Pairwise sequence alignments show that the ten 23S rRNA sequences are highly conserved (**Figure R3.2A**), and that mapping to rRNA is strongly hindered in uniquely mapping mode (**Figure R3.2B**), making it impossible to detect reliable cleavage sites in rRNA. In **Fig. R3.2B**, it is also evident that the proportion of rRNA reads in the WT and $\Delta ymcA$ strains is comparable, indicating similar rRNA depletion efficiency.

Figure R3.2 A) Pairwise nucleotide identity of the 10 copies of 23S rRNA. **B)** Boxplot of biotype distribution comparing multi- vs uniquely mapping reads in each strain. Data is from the late exponential phase.

Based on these results, we decided to validate our detected cleavages in 23S rRNAs using two different approaches: aligning the reads **A)** uniquely to each of the ten 23S rRNA sequences and **B)** to the 23S rRNA consensus sequence (from RNA central database). Both approaches

confirmed the sites we had identified, with the exception of a single site (position 1141), which we removed from our list of Y-complex-dependent cleavage sites. As a result, we have updated the materials and methods (L590-594) and **Supplementary Figs 5 & 6**, and improved the visualization of ribosomes in **Supplementary Fig 6**.

4. While the RNA processing data suggest the Y-complex guides RNase Y to specific cleavage sites, the precise molecular mechanism of substrate recognition remains somewhat speculative. The evidence for initiation of rRNA decay includes Y-complex-dependent cleavage sites within 23S and 16S rRNA (Figure 3A; Supplementary Figures 5A-B and 6), reduced 23S/16S degradation intermediates in the ymcA deletion (Figure 3B; Supplementary Figure 7A), and recovery of RNase Y together with ribosomal proteins in a YaaT-FLAG pulldown (Supplementary Figure 7B). These observations coherently support rRNA decay guided by the Y-complex and are persuasive. However, the catalytic role of RNase Y in executing these rRNA cleavages is inferred rather than directly demonstrated. The study does not directly test RNase Y catalytic dependency for the observed rRNA cleavages and ribosome decay in vivo. Attribution of rRNA cleavage to RNase Y rests on the loss of Y-complex-dependent ends and associations in Supplementary Figure 7B, but no RNase Y catalytic mutant, conditional depletion, or acute inhibition is used to show loss of the cleavage bands seen in Figure 3B and Supplementary Figure 7A. Without a direct RNase Y dependency test, the cleavages mapped in Figure 3A and the pulldown associations could be consistent with other endonucleases initiating decay, leaving the initiating nuclease unresolved.

We thank the reviewer for the insightful suggestion. We agree that providing direct evidence of RNase Y involvement in rRNA decay strengthens the mechanism we propose.

To address this, we constructed an RNase Y conditional strain to evaluate whether the observed rRNA cleavages depend on this endonuclease. For this, we inserted a copy of *rny* (gene encoding RNase Y) under the control of a xylose-inducible promoter (P_{xyIA}) in the chromosomal *lacA* locus. We then deleted the original copy of *rny*. In this way, the transcription of *rny* is entirely dependent on the addition of xylose to the media.

Depletion of RNase Y results in a significant growth defect in the transition phase (**Fig. R3.3A**), similar to the phenotype of the Y-complex mutants. However, the magnitude of the defect is stronger, consistent with RNase Y's role as the main endonuclease of *B. subtilis*. Importantly, the growth defect was fully rescued by xylose addition, confirming that the observed phenotype is due to reduced RNase Y levels.

Next, we determined the presence of 23S rRNA degradation intermediates in the RNase Y-depleted strain during the late exponential phase using Northern blot analysis. The depletion of RNase Y strongly reduces the accumulation of degradation intermediates (**Fig. R3.3B**), confirming its role in rRNA degradation.

Figure R3.3 Depletion of RNase Y results in a growth defect and reduces the degradation of ribosomes. A) Growth curve of RNase Y conditional depletion strains in LB. Expression of RNase Y depends on the addition of xylose to the media. **B)** Northern blot against 23S of WT and RNase Y-depleted cells at the late exponential phase. The methylene blue-stained membrane is shown as a loading control. This image is representative of three biological replicates

These new experimental findings, together with literature demonstrating the interaction between the Y-complex and RNase Y in RNA processing, provide direct evidence for the involvement of the Y-complex and RNase Y in rRNA decay.

We have included these results in **Supplementary Fig. 7C-D** and have written in lines L230-236: “To confirm the involvement of RNase Y in rRNA degradation, we constructed a conditional depletion strain in which the only copy of *rny* (coding for RNase Y) is under the control of a xylose-inducible promoter. Depletion of RNase Y results in a growth defect during the transition phase, mirroring what we observed with Y-complex mutants (**Supplementary Fig. 7C**). The degradation of 23S rRNA is highly impaired by the depletion of RNase Y (**Supplementary Fig. 7D**), confirming the involvement of RNase Y together with the Y-complex in ribosome degradation.”

5. Ribosome abundance changes are inferred from proteomics showing reduced ribosomal protein levels in wild type but impaired reduction in the *ymcA* deletion during transition (Figure 3C), unchanged rRNA and r-protein promoter activities in both strains (Figure 3D-E; Supplementary Figures 7C-E), and sedimentation profiles indicating 100S accumulation in the *ymcA* deletion (Figure 3F). These data strongly suggest altered ribosome homeostasis linked to Y-complex function. However, they do not explicitly separate decay of existing ribosomes from changes in biogenesis and assembly dynamics. The experiments do not directly quantify ribosome turnover rates to distinguish active decay of mature ribosomes from altered biogenesis or assembly states. Promoter assays in Figure 3D-E and Supplementary Figures 7C-E address transcriptional control but not rRNA synthesis rates, assembly efficiency, or ribosome half-lives. Proteomic mass fractions in Figure 3C reflect net abundance but cannot resolve whether changes arise from degradation versus assembly alterations; sedimentation (Figure 3F) indicates sequestration into 100S but does not quantify 70S turnover. Without pulse-chase labeling of rRNA or ribosome-specific turnover assessments, attributing abundance changes primarily to Y-complex-initiated decay remains an inference.

We thank the reviewer for their constructive feedback and experimental suggestion. We agree that we were missing a direct measurement of ribosome turnover to account for differences in the dynamics of ribosome biogenesis and assembly between WT and $\Delta ymcA$.

To address this, we have developed a fluorescence microscopy-based pulse-chase assay that utilizes the enzymatic properties of HaloTag to monitor ribosomal half-life *in vivo*.

We constructed an IPTG-inducible L1-HaloTag fusion protein that was integrated into the *amyE* locus. The L1 protein has been successfully used to track the ribosomes of *E. coli* and *B. subtilis* (Nikolay *et al.* 2015; Stoll *et al.* 2022). We labeled ribosomes during the exponential phase by inducing L1-HaloTag expression and adding the cell-permeable fluorescent TMR HaloTag ligand. The TMR ligand will covalently bind to the L1-HaloTag in the cell, enabling visualization and quantification by fluorescence microscopy.

When cells reached the late exponential phase (OD ~ 1), we exchanged the media for conditioned media obtained from cultures grown to the same cell density, and blocked further labelling by adding the non-fluorescent HaloTag blocker 7-bromo-1-heptanol (Merrill *et al.* 2019). We quantified ribosome abundance and decay by measuring bacterial fluorescence under a microscope (**Figure R3.4A**). We use conditioned media for the chase to mimic the nutrient availability during the transition phase and avoid introducing new nutrients by using fresh media. The deletion of *ymcA* significantly reduces the decay of labeled ribosomes, duplicating the half-life calculated for the WT strain (22.8 min for WT and 51.2 min for $\Delta ymcA$) (**Figure R3.4B**). These results provide direct evidence that the Y-complex is required for active ribosome degradation during the transition phase.

Figure R3.4 Ribosome decay at the late exponential phase. A) Rationale of pulse-chase experiment. **B)** Decay of labeled ribosomes after the exponential phase. The mean fluorescence intensity (MFI) per cell was measured by epifluorescence microscopy and normalized to the MFI at timepoint 0. Around 200 cells were analyzed by time point and strain, representing two independent experiments. The boxplot represents the interquartile range (IQR) and the median in the center. Whiskers show the variability outside quartile 1 (Q1) and Q3 and were calculated as $Q1-1.5 \cdot IQR$ and $Q3+1.5 \cdot IQR$, respectively. The statistical significance was tested using Wilcoxon test. ****: p -value ≤ 0.0001 . The half-life ($t_{1/2}$) was

calculated by fitting a one-phase decay function. The data represent the $t_{1/2}$ average \pm S.D of two independent experiments.

We have added these results in **Figure 3F-G** and have written the following in lines L255-264:

“To distinguish active ribosome degradation from changes in their biogenesis, we quantified ribosome turnover *in vivo* using an inducible uL1-HaloTag fusion to label the 50S subunit. Ribosomes were pulse-labelled with a fluorescent TMR-ligand during the exponential phase and subsequently chased in conditioned media after blocking further labelling with 7-bromo-1-heptanol⁴² and analyzed by microscopy (**Fig. 3F**). Ribosome half-life more than doubled in the $\Delta ymcA$ mutant (51.2 ± 10.7 min) compared to the wild-type (22.8 ± 1.08 min) during the transition to stationary phase (**Fig. 3G**). These results suggest that the Y-complex is required for the active turnover of mature ribosomes, confirming that the observed accumulation of ribosomes in the $\Delta ymcA$ is driven by reduced degradation rather than increased ribosome biogenesis.”

References:

Merrill, R. A., Song, J., Kephart, R. A., Klomp, A. J., Noack, C. E., & Strack, S. (2019). A robust and economical pulse-chase protocol to measure the turnover of HaloTag fusion proteins. *Journal of Biological Chemistry*, 294(44), 16164-16171.

Nikolay, R., Schloemer, R., Mueller, S., & Deuerling, E. (2015). Fluorescence-based monitoring of ribosome assembly landscapes. *BMC Molecular Biology*, 16(1), 3.

Stoll, J., Zegarra, V., Bange, G., & Graumann, P. L. (2022). Single-molecule dynamics suggest that ribosomes assemble at sites of translation in *Bacillus subtilis*. *Frontiers in Microbiology*, 13, 999176.

6. Support for functional specificity rests on the overrepresentation analysis showing enrichment of Y-complex-dependent cleavages in metabolism and information-processing categories (Figure 2G; Supplementary Figure 4A). In the same dataset, there is a clear preference for cleavages in polycistronic transcripts and internal UTRs (Figure 2E–F; Supplementary Figure 2C). The authors also show that transcript length and abundance alone do not predict cleavage (Supplementary Figure 2D–E), which is reassuring. The enrichment analysis does not control for operon architecture and other structural covariates that could confound the apparent functional specificity. Because the data show a pronounced bias toward polycistronic mRNAs (Figure 2E) and multiple cleavages per polycistronic transcript (Supplementary Figure 2C), functional categories that are predominantly operonic (such as metabolic and translation operons) are more likely to appear enriched in Figure 2G. Although length and abundance were examined (Supplementary Figure 2D–E), the analysis does not stratify by polycistronic versus monocistronic organization, UTR frequency, motif density, or local folding energy, leaving open the possibility that the observed enrichment reflects architecture rather than function. Without a multivariate or stratified enrichment framework controlling for these structural covariates, the conclusion of functional specificity remains partially confounded by transcript architecture.

We agree with the reviewer that the specificity of the Y-complex is likely driven by structural features and transcript architecture rather than the function of the encoded gene itself. We believe that the results of our enrichment analysis reflect an evolutionary outcome in which transcripts that acquired the specific structural elements required for Y-complex-mediated

processing were selected for because of the fitness advantages they confer during growth phase transitions. Specifically, the cleavage and subsequent post-transcriptional regulation of transcripts encoding key metabolic and central functions are likely highly advantageous when cells must rapidly adapt to changing environmental conditions.

As the reviewer noted, we observe a bias toward polycistronic mRNAs, which likely points toward the known role of the Y-complex in transcript maturation and the generation of new isoforms. To directly address and mitigate the potential bias introduced by polycistronic organization, we conducted our overrepresentation analysis (ORA) using operons as the fundamental functional unit rather than individual genes. By comparing the number of targeted operons against the total number of operons within each functional category, we explicitly account for the clustering of metabolic and informational genes. This methodology ensures that a single large operon containing one or multiple cleavage sites does not disproportionately overestimate the significance of any given functional category.

Our objective was not to assert that the functional category determines the cleavage event, but rather to identify whether the targeted mRNAs are associated with specific, biologically relevant pathways. In support of this functional relevance, we observe that the categories enriched for Y-complex targets also exhibit significant changes in overall gene expression (**Supplementary Fig. 4B**).

To clarify this perspective in the manuscript, we have revised the text in lines L184-189 as follows: “A specific cleavage motif of the Y-complex could not be identified. Therefore, we wondered whether this multifactorial cleavage motif has evolved in transcripts with specific functions required to switch the growth phase. We conducted an operon-based overrepresentation analysis to determine whether the target transcripts of the Y-complex encode genes with specific functions. We observed that the most significantly enriched categories pertain to “Metabolism” and “Information processing” (**Fig. 2G, Supplementary Fig. 4A, Supplementary Table 5**).”